# Periodic propagating waves coordinate RhoGTPase network dynamics at the leading and trailing edges during cell migration

Alfonso Bolado-Carrancio[1†], Oleksii S Rukhlenko[2†], Elena Nikonova[2], Mikhail A Tsyganov[2,3], Anne Wheeler[1], Amaya Garcia-Munoz[2], Walter Kolch[2,4], Alex von Kriegsheim[1,2]*, Boris N Kholodenko[2,4,5]*

[1]Edinburgh Cancer Research Centre, Institute of Genetics and Molecular Medicine, University of Edinburgh, Edinburgh, United Kingdom; [2]Systems Biology Ireland, School of Medicine and Medical Science, University College Dublin, Belfield, Ireland; [3]Institute of Theoretical and Experimental Biophysics, Pushchino, Russian Federation; [4]Conway Institute of Biomolecular & Biomedical Research, University College Dublin, Belfield, Ireland; [5]Department of Pharmacology, Yale University School of Medicine, New Haven, United States

**Abstract** Migrating cells need to coordinate distinct leading and trailing edge dynamics but the underlying mechanisms are unclear. Here, we combine experiments and mathematical modeling to elaborate the minimal autonomous biochemical machinery necessary and sufficient for this dynamic coordination and cell movement. RhoA activates Rac1 via DIA and inhibits Rac1 via ROCK, while Rac1 inhibits RhoA through PAK. Our data suggest that in motile, polarized cells, RhoA–ROCK interactions prevail at the rear, whereas RhoA-DIA interactions dominate at the front where Rac1/Rho oscillations drive protrusions and retractions. At the rear, high RhoA and low Rac1 activities are maintained until a wave of oscillatory GTPase activities from the cell front reaches the rear, inducing transient GTPase oscillations and RhoA activity spikes. After the rear retracts, the initial GTPase pattern resumes. Our findings show how periodic, propagating GTPase waves coordinate distinct GTPase patterns at the leading and trailing edge dynamics in moving cells.

**\*For correspondence:**
Alex.VonKriegsheim@igmm.ed.ac.uk (AK);
boris.kholodenko@ucd.ie (BNK)

[†]These authors contributed equally to this work

**Competing interests:** The authors declare that no competing interests exist.

## Introduction

Cell migration relies on the coordination of actin dynamics at the leading and the trailing edges (*Ridley et al., 2003*). During the mesenchymal type of migration, protrusive filamentous actin (F-actin) is cyclically polymerized/depolymerized at the cell's leading edge, whereas the contractile, actomyosin-enriched trailing edge forms the rear. The leading edge protrudes and retracts multiple times, until the protrusions, known as lamellipodia, are stabilized by adhering to the extracellular matrix (*Ridley, 2001*). Subsequently, the cell rear detaches and contracts allowing the cell body to be pulled toward the front. Core biochemical mechanisms of this dynamic cycle are governed by the Rho family of small GTPases (*Jaffe and Hall, 2005*). Two members of this family, Ras homolog family member A (RhoA) and Ras-related C3 botulinum toxin substrate 1 (Rac1), control protrusions and retractions at the leading edge as well as the contractility at the rear (*Felmet et al., 2005*; *Heasman and Ridley, 2008*; *Machacek et al., 2009*). RhoGTPases cycle between an active, GTP-loaded 'on' state and an inactive, GDP-loaded 'off' state. Switches between on and off states are tightly regulated by (i) guanine nucleotide exchange factors (GEFs) that facilitate GDP/GTP

exchange thereby activating GTPases and (ii) GTPase activating proteins (GAPs) that stimulate GTP hydrolysis and transition to a GDP-bound state.

A canonic description of mesenchymal cell migration portrays mutually separated zones of Rac1-GTP and RhoA-GTP in polarized cells where Rac1-GTP dominates at the leading edge and RhoA-GTP dominates at the contracted cell rear (*Holmes and Edelstein-Keshet, 2016*; *Holmes et al., 2017*; *Kunida et al., 2012*; *Kurokawa and Matsuda, 2005*; *Pertz et al., 2006*; *Wang et al., 2013*; *Zmurchok and Holmes, 2020*). This distinct distribution of RhoA and Rac1 activities along polarized cells is explained by a mutual antagonism of RhoA and Rac1 (*Edelstein-Keshet et al., 2013*; *Mori et al., 2008*) mediated by downstream effectors of these GTPases (*Byrne et al., 2016*; *Guilluy et al., 2011a*; *Pertz, 2010*). The Rac1 effector, p21 associated kinase (PAK), phosphorylates and inhibits multiple RhoA-specific GEFs, including p115-RhoGEF, GEF-H1 and Net1 (*Alberts et al., 2005*; *Guilluy et al., 2011a*; *Rosenfeldt et al., 2006*). In addition, active Rac1 binds and activates p190RhoGAP, which decreases RhoA activity (*Guilluy et al., 2011a*). In turn, RhoA-GTP recruits the Rho-associated kinase (ROCK), which phosphorylates and activates Rac-specific GAPs, such as Fil-GAP and ArhGAP22, thereby inhibiting Rac1 (*Guilluy et al., 2011a*; *Ohta et al., 2006*; *Sanz-Moreno et al., 2008*). This mutual inhibition of RhoA and Rac1 may lead to a bistable behavior where a system can switch between two stable steady states, in which GTPase activities alternate between high and low values (*Kholodenko, 2006*; *Mori et al., 2008*). The existence of bistable switches is supported by experiments, where inhibition of the Rac1 effector PAK maintains both high RhoA and low Rac1 activities and associated morphological changes even after the inhibition is released (*Byrne et al., 2016*).

At the same time, RhoA and Rac1 do not behave antagonistically at the leading edge of migrating cells. Here, RhoA activation is rapidly followed by Rac1 activation, tracking a protrusion-retraction cycle (*Machacek et al., 2009*). This Rac1 activation at the leading edge is mediated by the downstream RhoA effector, Diaphanous related formin-1 (DIA), that was shown to localize to the membrane ruffles of motile cells (*Tkachenko et al., 2011*; *Watanabe et al., 1997*). Thus, in contrast to the RhoA effector ROCK, which inhibits Rac1 in the other cell segments, the RhoA effector DIA can stimulate Rac1 activity at the leading edge.

If at the leading edge RhoA activates Rac1 but Rac1 inhibits RhoA, this intertwined network circuitry of positive and negative loops will force the network to periodically change RhoA and Rac1 activities, giving rise to self-perpetuating oscillations with a constant amplitude and frequency (*Kholodenko, 2006*; *Tsyganov et al., 2012*). By contrast, at the trailing edge and cell body, the mutual RhoA and Rac1 inhibition results in the maintenance of a (quasi)steady state with high RhoA activity and low Rac1 activity. But, how can these different dynamics coexist? More importantly, how are these dynamics coordinated within the cell? Despite decades of research that have painstakingly characterized dynamic Rho and Rac behaviors in cell motility (*Holmes and Edelstein-Keshet, 2012*), we do not know what dynamic features are necessary and sufficient to achieve the biological effect of cell motility, and how different dynamics at the front and rear are coordinated.

Here, we first elucidated the spatial profiles of RhoA-Rac1 interactions in motile MDA-MB-231 breast cancer cells. Using proximity ligation assays (PLA), we show that the concentration of complexes formed by RhoA and its downstream effectors DIA and ROCK depends on the spatial location along the longitudinal axis of polarized cells. RhoA primarily interacts with DIA at the cell leading edge, whereas RhoA - ROCK interactions are the strongest at the cell rear. Based on these findings, we built a mathematical model to analyze RhoA-Rac1 signaling in space and time. The model predicts and the experiments corroborate that at the cell front the GTPase network exhibits oscillatory behavior with high average Rac1-GTP, whereas at the cell rear there is a (quasi)steady state with high RhoA-GTP and low Rac. The front and rear are connected by periodic, propagating GTPase waves. When the wave reaches the rear, RhoA-GTP transiently oscillates and then, following the rear retraction, the GTPase network dynamic pattern returns to the original state. Our model and experimental results show how different GTPase dynamics at the leading edge and the trailing edge can govern distinct cytoskeleton processes and how moving cells reconcile these different dynamics. The RhoA-Rac1 interaction network model defines minimal, autonomous biochemical machinery that is necessary and sufficient for biologically observed modes of cell movement.

## Results

### Spatially variable topology of the RhoA-Rac1 interaction network

The Rac1 effector PAK inhibits RhoA, and the RhoA effector ROCK inhibits Rac1 (*Guilluy et al., 2011a*). Here, we tested how the other RhoA effector, DIA, influences the Rac1 and RhoA activities. We first downregulated DIA using small interfering RNA (siRNA) and measured the resulting changes in the Rac1-GTP and RhoA-GTP levels. Downregulation of DIA increased the RhoA abundance and decreased Rac1 abundance, while decreasing relative activities of both RhoA and Rac1 (*Figure 1—figure supplement 1*, panels A and B). The decrease of relative Rac1 and RhoA activities induced by DIA knockdown shows that DIA activates Rac1 and also supports the existence of a positive feedback loop between DIA and RhoA described earlier (*Kitzing et al., 2007*). In addition, the GTPase network features another positive feedback from PAK to Rac1 through several molecular mechanisms (*Baird et al., 2005*; *DerMardirossian et al., 2004*; *Feng et al., 2002*; *Obermeier et al., 1998*). Summing up the interactions between RhoA and Rac1 mediated by their effectors ROCK and PAK (*Byrne et al., 2016*) and RhoA - Rac1 interactions through DIA, we arrive at the intertwined negative and positive feedback circuitry of the RhoA-Rac1 network shown in *Figure 1—figure supplement 1*, panel C.

To explain the distinct GTPase activities at the leading and trailing edges, we hypothesized that these diverse feedforward and feedback mechanisms may be spatially controlled. Therefore, we explored how the interactions of active RhoA with its effectors vary spatially in polarized MDA-MB-231 cells. Using a proximity ligation assay (PLA), which visualizes protein interactions in situ (*Gustafsdottir et al., 2005*; *Söderberg et al., 2006*), we measured RhoA-DIA and RhoA-ROCK complexes (*Figure 1A and B*). Based on the commonly considered morphology of the long, narrow cell rear and the wide leading edge (*Caswell and Zech, 2018*), we segmented each polarized cell into three parts: the rear (about 20% of the cell length), intermediate region (next 70% of the cell length), and front (the rest 10% of the length). The density of the RhoA-effector complexes was quantified by dividing the number of PLA reactions by the area of the corresponding compartment.

The results show that the RhoA-DIA complexes are predominantly localized at the cell front, whereas their density is markedly decreased at the rear (*Figure 1A*). In contrast, the density of the RhoA-ROCK complexes increases toward the cell rear and decreases at the leading edge (*Figure 1B*). These results are in line with protein staining data in polarized cells, which suggest that DIA is mainly localized at the leading edge (*Figure 1C*), whereas ROCK is abundant at the rear and cell body (*Figure 1D*; *Brandt et al., 2007*; *Goulimari et al., 2005*; *Newell-Litwa et al., 2015*; *Watanabe et al., 1997*; *Wheeler and Ridley, 2004*). For MDA-MB-231 cells, our quantitative proteomics data showed that the RhoA abundance is at least 10-fold larger than the abundance of DIA and ROCK isoforms combined (*Byrne et al., 2016*). Thus, as shown in the Modeling section of Materials and methods, the RhoA-effector concentrations depend approximately linearly on the DIA and ROCK abundances. Taken together, these results suggest a protein interaction circuitry of the GTPase network, where competing effector interactions are spatially controlled (*Figure 1E*). In order to analyze how this differential spatial arrangement of GTPase-effector interactions can accomplish the dynamic coordination between the leading and trailing edges, we constructed a mechanistic mathematical model and populated it by quantitative mass spectrometry data on protein abundances (*Supplementary file 1*).

### Analyzing the dynamics of the RhoA-Rac1 interaction network

The changes in ROCK and DIA abundances along the longitudinal axis of polarized cells (*Figure 1C and D*) could plausibly encode the distinct RhoA-Rac1 temporal behaviors in different cellular segments. Therefore, we explored these possible dynamics of the GTPase network for different DIA and ROCK abundances prevailing at different spatial positions along the cell length. We first used a spatially localized, compartmentalized model where different DIA and ROCK abundances corresponded to distinct spatial locations (see Modeling section of Materials and methods for a detailed description of this model).

Using the model, we partitioned a plane of the ROCK and DIA abundances into the areas of different temporal dynamics of RhoA and Rac1 activities (*Figure 2A*). This partitioning is a two-parameter bifurcation diagram where the regions of distinct GTPase dynamics are separated by bifurcation

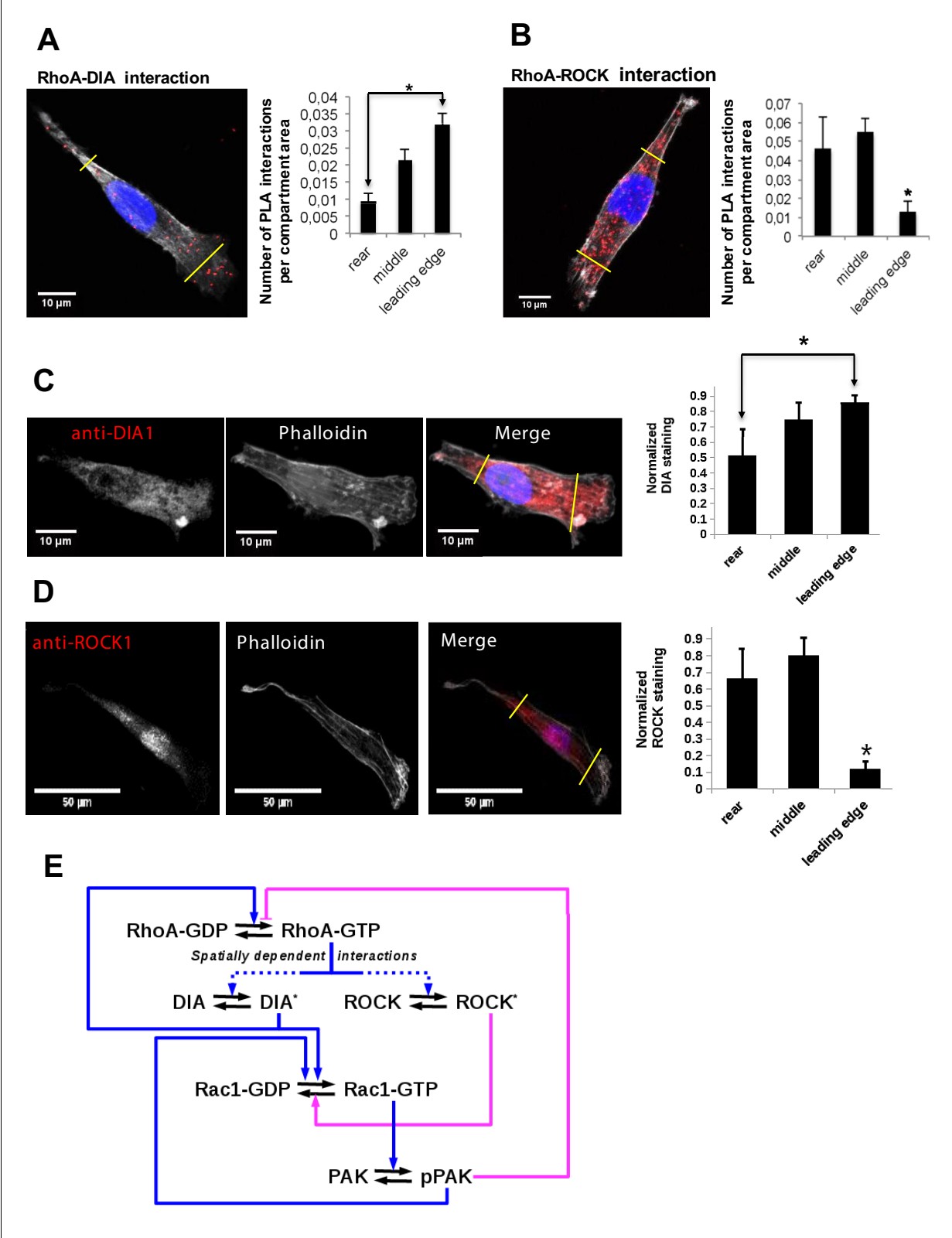

**Figure 1.** Differential localization of the RhoA-DIA and RhoA-ROCK1 protein complexes determine spatially resolved signaling topology. (A, B) Representative PLA images. Each red spot within a cell represents a fluorescent signal from a single RhoA-DIA1 (A) or RhoA-ROCK1 (B) complex. Yellow lines indicate bounds for the leading edge, intermediate region and rear. Bar graphs at the right show the average density of these complexes in different cell regions (the rear, middle and leading edge)± S.E.M. of four independent experiments with 25 cells analyzed per experiment. The

*Figure 1 continued on next page*

*Figure 1 continued*

asterisk * indicates that p<0.05 calculated using unpaired t-test. (**C, D**) Representative images of DIA1 and ROCK1 immunostaining. Bar graphs at the right show quantified immunostaining density signals for different cellular compartments ± S.E.M. of four independent experiments with one cell analyzed per experiment. The asterisk * indicates that p<0.05 calculated using unpaired t-test. (**E**) A schematic wiring diagram of the RhoA-Rac1 network, showing positive (blue) and negative (magenta) feedback loops. Spatially varying RhoA interactions with its effectors DIA and ROCK are shown by dashed lines.

The online version of this article includes the following figure supplement(s) for figure 1:

**Figure supplement 1.** Elucidation of the topology of the RhoA GTPase network: DIA knockdown influence on the GTPase activities.

boundaries at which abrupt, dramatic changes in the dynamic behavior occur (*Holmes and Edelstein-Keshet, 2016*). The blue region 1 in *Figure 2A* corresponds to the self-perpetuating oscillations of the RhoA and Rac1 activities at the leading edge. The ROCK abundance is markedly lower and the DIA abundance is higher at the leading edge than in the cell body (*Figure 1C and D*). Thus, a combination of Rac1 activation by RhoA via DIA and RhoA inhibition by Rac1 via PAK (*Figure 2B*) results in sustained oscillations of RhoA and Rac1 activities at the leading edge (*Figure 2D*). This periodic Rac1 activation drives actin polymerization at the leading edge pushing protrusion-retraction cycles (*Machacek et al., 2009*; *Martin et al., 2016*; *Pertz, 2010*; *Tkachenko et al., 2011*).

The green region 2 in *Figure 2A* is an area of stable high RhoA and low Rac1 activities at the rear and intermediate cell regions. Within this region, RhoA inhibits Rac1 via ROCK, and Rac1 inhibits RhoA via PAK (*Figure 2C*). After perturbations, the GTPase network converges to steady-state levels of high RhoA-GTP and particularly low Rac1-GTP (*Figure 2E*). Unlike other dynamical regimes with only a single stable steady state, region 2 corresponds to an excitable an medium, which cannot generate pulses itself, but supports the propagation of excitable activity pulses (see Materials and methods section).

The red region 3 corresponds to the coexistence of GTPase oscillations and a stable steady state with high RhoA and low Rac1 activities. Depending on the initial state, the GTPase network evolves to different dynamic regimes. If the initial state has high RhoA-GTP and low Rac1-GTP, the GTPase network progresses to a stable steady state, but if the initial state has low RhoA-GTP and high Rac1-GTP, the network will develop sustained oscillations (*Figure 2F*). This region 3 is termed a BiDR (<u>Bi</u>-<u>D</u>ynamic-<u>R</u>egimes) by analogy with a <u>bi</u>-<u>s</u>table region where two stable steady states coexist and the system can evolve to any of these states depending on the initial state (*Kholodenko, 2006*). However, in contrast with bistable regimes only one of two stable regimes is a stable steady state in the BiDR region, whereas the other dynamic regime is a limit cycle that generates stable oscillations.

In addition to these dynamic regimes, the spatially localized model predicts other emergent nonlinear dynamic behaviors (*Figure 2A*, *Figure 2—figure supplement 1*, panels A-D, and *Figure 2—figure supplement 2*), which the GTPase network may execute under large perturbations of the RhoA and Rac1 effector abundances to coordinate GTPase signaling at the leading and trailing edges (see Modeling section of Materials and methods for a detail description of these regimes). Therefore, we next analyzed how the leading and trailing edge GTPase dynamics are coupled.

## Spatiotemporal dynamics of the RhoA-Rac1 network reconciles the distinct temporal behaviors at the cell front and rear

Different active GTPase concentrations in the cell rear and the leading edge induce diffusion fluxes (*Das et al., 2015*), which in turn influence the emerging behavior of these GTPases and coordinate their dynamics in distinct cellular segments. As a multitude of dynamic behaviors is possible, we systematically explored the behavior of the RhoA-Rac1 network in space and time using a spatiotemporal model of the GTPase network interactions (referred to as a reaction-diffusion model, see Materials and methods). Starting from experimental observations to rationalize which behaviors are likely with physiological boundaries, we digitized 2D images of polarized cells and incorporated the DIA and ROCK abundances as functions of the spatial coordinate along the cell length, based on the quantitative imaging data (*Figure 3A–C*).

The model predicts autonomous, repeating cycles of the spatiotemporal GTPase dynamics (*Figure 3D–G* and *Video 1*). For a substantial part of a dynamic cycle, high RhoA-GTP and low Rac1-GTP persist at the cell rear and maintain the rear contraction, whereas active RhoA and Rac1

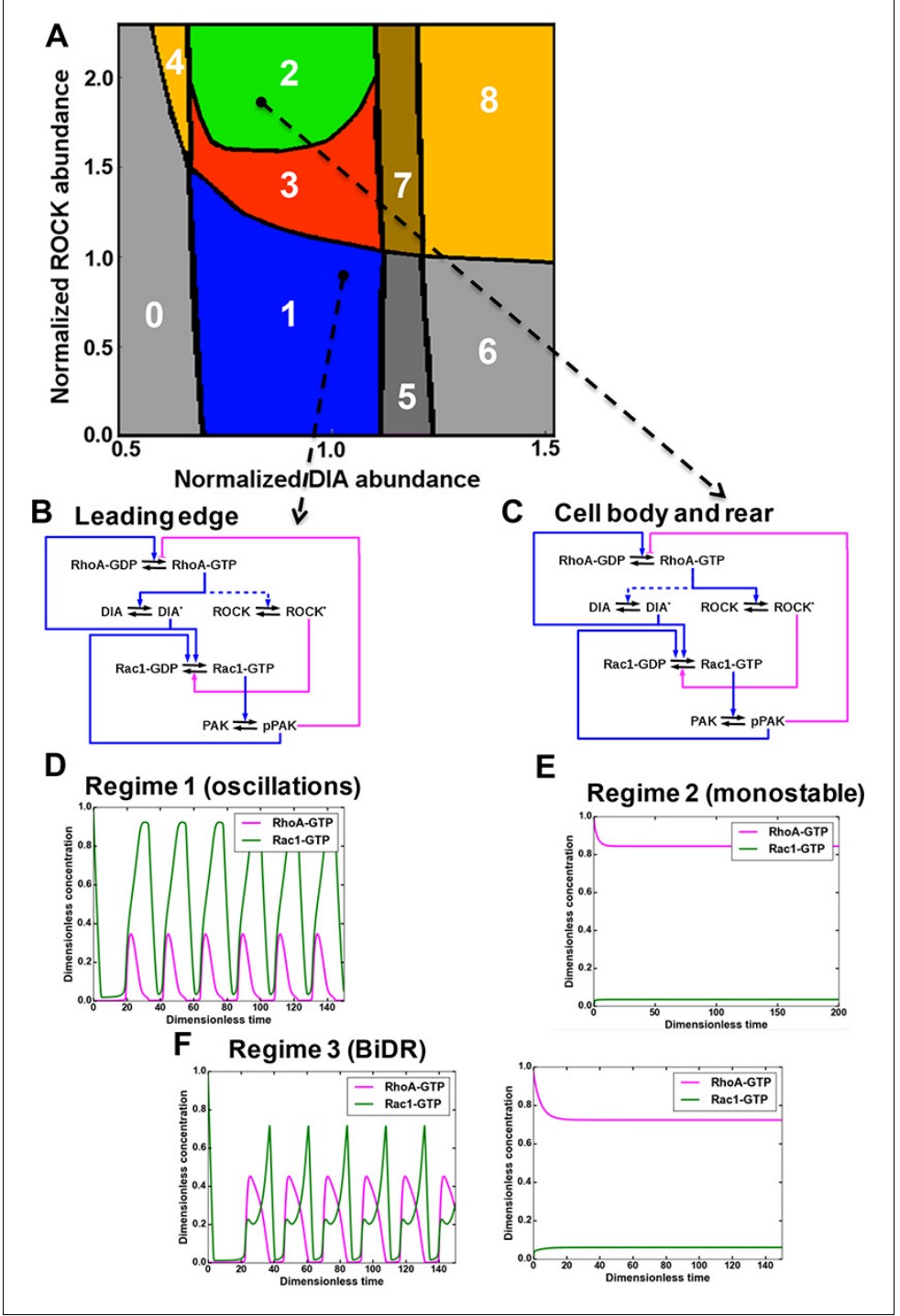

**Figure 2.** A mathematical model of the RhoA-Rac1 network predicts dramatically distinct dynamic regimes for different DIA and ROCK abundances. (**A**) Distinct dynamic regimes of the RhoA-Rac1 network dynamics for different DIA and ROCK abundances. Oscillations of RhoA and Rac1 activity exist within area 1 (regime 1). In area 3, sustained GTPase oscillations and a stable steady state with high RhoA and low Rac1 activities coexist. Regimes 0, 2, 5 and 6 have only one stable steady state. Notably, regime 2 is excitable. Steady state solutions with high RhoA activity exist in areas 2–4, and 6–8. Stable steady state solutions with high Rac1 activity exist in areas 0 and 5–8. Regimes 4, 7 and 8 are bistable with two stable steady states. (**B, C**) Wiring diagrams of the RhoA-Rac1 network for the cell leading edge (**B**) and the cell body and rear (**C**). Dashed blue lines indicate weak activating connections. (**D–F**) Typical time courses of RhoA and Rac1 activity in regimes 1 (**D**), and 2 (**E**). (**F**) In area 3,

*Figure 2 continued on next page*

*Figure 2 continued*

depending on the initial state, the GTPase network evolves either to a stable steady state (right) or a stable oscillatory regime (left).

The online version of this article includes the following figure supplement(s) for figure 2:

**Figure supplement 1.** Distinct dynamic regimes of the RhoA-Rac1 network for different effector abundances.

**Figure supplement 2.** Nullclines and vector fields describing the nine dynamic regimes of RhoA-GTP and Rac1-GTP shown in *Figure 2A*.

**Figure supplement 3.** One-parameter bifurcation diagrams for changing ROCK and DIA abundances separately in *Figure 2A*.

oscillate at the leading edge, resulting in actin (de)polarization cycles and protrusion-retraction cycles (*Figure 3D and F*; *Wang et al., 2013*). At the same time, a wave of oscillating Rac1 and RhoA activities slowly propagates from the leading edge toward the cell rear (*Figure 3E and G*). Between the oscillatory RhoA-GTP zone and the areas of high RhoA activity, a zone of low RhoA activity emerges (*Figure 3F*). As time progresses, the wave of oscillating GTPase activities and the area of low RhoA activity spread to the rear (*Figure 3—figure supplement 1*, panels A and B), leading to re-arrangement of the cytoskeleton (*Warner et al., 2019*). Because of the oscillations, zones of low Rac1 activities emerge, which give rise to high RhoA-GTP that interacts with ROCK and leads to the rear retraction (*Video 1*). Subsequently, RhoA returns to its initial high stable activity, and the dynamic pattern of RhoA-GTP and Rac1-GTP over the entire cell returns to its initial state. These model simulations could plausibly explain how the different GTPase dynamics at the cell front and rear are coordinated to enable successful cell migration.

Therefore, it was important to test the prediction arising from the model simulations in biological experiments. For this, we used cells stably expressing the mTFP-YFP RhoA-GTP FRET-probe (*Kim et al., 2015*) allowing us to determine the RhoA-GTP dynamics using ratiometric, live-cell spinning disk microscopy. We imaged the cells with a frequency of one image every 5 s and constrained the measurement time to 10 min to limit phototoxic effects. Due to this time limitation, a full cycle of cellular movement (around 45 min on average, *Video 2*) could not be followed in an individual cell, and the full spatiotemporal RhoA activity cycle during a cell movement was compiled from several cells observed in different phases of cellular movement. In the initial phase of the cell movement cycle, the spatiotemporal RhoA activity showed three different zones: (*i*) oscillations at the leading edge, (*ii*) dark zone of low activity and (*iii*) light zone of high activity (*Figure 3H* and *Figure 3—figure supplement 1*, panel C) in the cell body and rear, matching the model prediction (Figsure 3F and *Figure 3—figure supplement 1*, panel C). As time progressed, the GTPase activity wave propagated further into the cell (*Figure 3I*), forming zones of high and low RhoA activities. In the space-time coordinates, the slope of the boundaries of these zones suggests that they travel from the leading edge to the cell rear, confirming the model predictions (*Video 1* and *Figure 3I*). When the wave of oscillatory GTPase activities finally reaches the cell rear, it induces several RhoA-GTP spikes (*Figure 3G and I*), periods of low RhoA activity (*Figure 3—figure supplement 1*, panels A-B and D), and subsequent return to the original, high RhoA-GTP at the rear and part of the cell body (*Figure 3F and H*). *Figure 3—figure supplement 1*, panel D experimentally captures this transition from a low RhoA activity to the original high activity as the final step of the cell movement cycle predicted by the model.

The model predicts that during a single cellular movement cycle, multiple bursts of RhoA activity appear at the leading edge, whereas at the cell rear, RhoA activity bursts occur only after the RhoA-Rac1 wave has spread through the cell (*Video 1*). Measuring the number of RhoA bursts at the leading edge and cell rear during observation time (10 min) corroborated model predictions, showing a ca. fivefold larger number of bursts at the leading edge than at the cell rear (*Figure 3J*). On average, at the leading edge a burst of RhoA activity happens every minute, while at the cell rear only 1 or 2 bursts happen during 10 min (*Figure 3J*).

Although spatially resolved Rac1 activity can be determined using exogenous probes, they dramatically change the cell shape when expressed (*Pertz, 2010*). However, endogenous Rac1-GTP can be reliably detected by immunostaining with a conformation-specific Rac1-GTP antibody. Rac1 was mainly active at the leading edge with lower activity in the space between the nucleus and cell rear (*Figure 3K*), similar to the patterns observed in the model for protrusion-retraction cycles

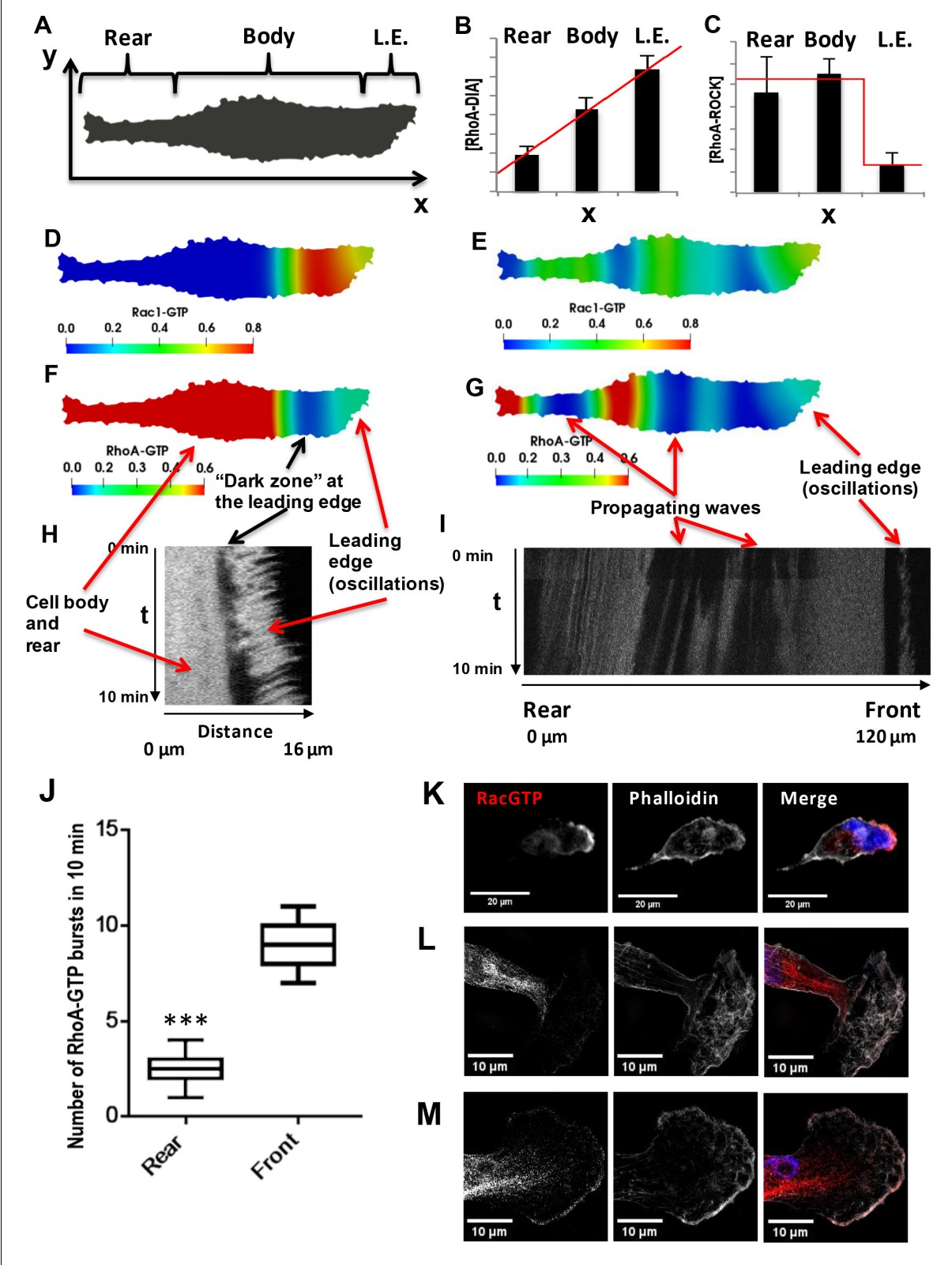

**Figure 3.** Spatial propagation of RhoA and Rac1 activities during cell motility. (**A**) A 2-D calculation domain obtained by digitizing cell images. Different cellular compartments are indicated. The x-axis represents the direction of cell polarization, the y-axis represents the perpendicular direction. (**B, C**) The abundance profiles of DIA and ROCK used in simulations (red lines) are superimposed on the experimental spatial profiles (bar graphs in **Figure 1C and D**). (**D–G**) Model-predicted spatial patterns of the RhoA and Rac1 activities for different phases of the cell movement cycle. (**D, F**) Rac1 and RhoA

*Figure 3 continued on next page*

*Figure 3 continued*

activity snapshots during a protrusion-retraction cycle at the leading edge (t = 175 s from the start of the moving cycle). (E, G) represent snapshots when the Rac1 and RhoA activity wave have spread over the entire cell, reaching the rear (t = 1518 s). (H) The RhoA activity at the leading edge and cell body during a protrusion-retraction phase measured by RhoA FRET probe in space and time. The arrows compare model-predicted and experimentally measured patterns, indicating zones of RhoA oscillatory and high constant activities and a 'dark zone' of low RhoA activity. (I) Spatiotemporal pattern of the RhoA activity during further RhoA wave propagation into the cell. (J) The number of RhoA activity bursts at the cell body and rear during 10 min measured using the RhoA FRET probe. Error bars represent $1^{st}$ and $3^{rd}$ quartiles, *** indicate p<0.001 calculated using unpaired t-test. (K–M) Fluorescent microscopy images of Rac1 activity (red), combined with staining for F-actin (phalloidin, white) and the nucleus (DAPI, blue) in fixed cells for different phases of the cell movement cycle; (K) a protrusion-retraction cycle at the leading edge, and (L, M) present Rac1 activity wave propagation into the cell body. The images (L, M) were obtained by super-resolution microscopy.

The online version of this article includes the following figure supplement(s) for figure 3:

**Figure supplement 1.** Spatial propagation of RhoA and Rac1 activities during cell motility.

---

(*Figure 3D*). The GTPase waves can be detected using super-resolution imaging. These images corroborated the Rac1-GTP presence towards the cell nucleus and rear (see super-resolution images in *Figure 3L–M* and *Figure 3—figure supplement 1*, panel E). The series of images shown in *Figure 3K–M* and *Figure 3—figure supplement 1*, panel E is consistent with the concept of traveling Rac1-GTP waves predicted by the model.

The spatiotemporal activation dynamics of Rac1 and RhoA underlie the morphological events during cell migration, that is protrusion-retraction cycles at the front and the retraction cycle at the rear (*Ridley et al., 2003*; *Video 2*). These mechanical processes, involving cytoskeleton proteins, can be coordinated by periodic propagating waves of RhoGTPase activities described by our model.

## Hysteresis of Rac1 and RhoA activities and cell shape features

We previously showed that PAK inhibition could change the cell shape of MDA-MB-231 cells from mesenchymal to amoeboid (*Byrne et al., 2016*). The mesenchymal mode of migration features an elongated cell morphology and high Rac1 activity, whereas the amoeboid mode is hallmarked by a rounded morphology and high RhoA activity (*Sanz-Moreno et al., 2008*). These morphologies and migration types are mutually exclusive but can transition into each other. Our previous study showed that this transition correlated with the hysteresis of active RhoA and Rac1 upon PAK inhibition (*Byrne et al., 2016*). Hysteresis is the hallmark of bistability: if a parameter, such as the PAK abundance, reaches a threshold value, then the system flips from one stable state to another stable state, at which it remains for a prolonged period of time even when this parameter has returned to its initial value (*Markevich et al., 2004*; *Sha et al., 2003*).

Our model now allows us to examine the exact spatiotemporal kinetics of the GTPase network in response to changes in PAK abundance or activity. Varying PAK causes Rac1 and RhoA activities to move through different dynamic regimes (shown by the line connecting points I – II – III in *Figure 4A*). In unperturbed cells, GTPase activities oscillate at the leading edge. This initial network state corresponds to point I in region 1n and unperturbed ROCK, PAK and DIA abundances and activities (the point I coordinates are (1, 1) in *Figure 4A*). Because Rac1 and RhoA are difficult to target for therapeutic interventions, we used a small molecule PAK inhibitor (IPA-3) in our previous study (*Byrne et al., 2016*). As PAK abundance gradually decreases (or PAK inhibition increases), the system moves from the oscillatory region 1 to the BiDR region 3, before reaching a bistable regime (regions 7 and 8), as shown by point II. In the BiDR region, (*i*) a stable high RhoA-GTP, low Rac1-GTP state and (*ii*) a stable oscillatory state with a high average Rac1-GTP coexist at the leading edge (*Figure 2F* and *Figure 2—figure supplement 2*, panel D). While

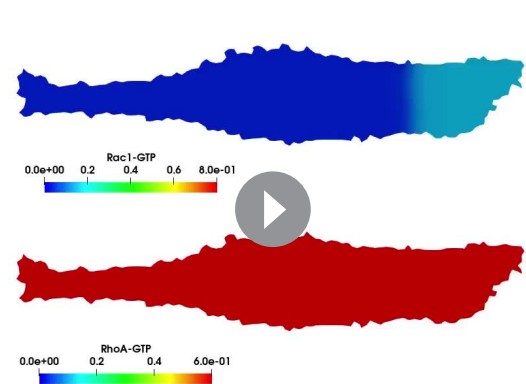

**Video 1.** Model-predicted spatiotemporal activity patterns of RhoA and Rac1.
https://elifesciences.org/articles/58165#video1

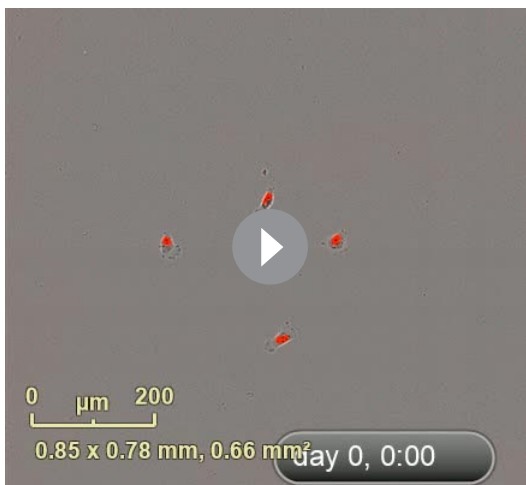

**Video 2.** Live-cell imaging of cel movement cycles. Red color represents staining of the nuclei. Frame increment is 10 min.
https://elifesciences.org/articles/58165#video2

moving from point I into area 3, the system continues to display the stable oscillatory state with high average Rac1-GTP. In the bistable regions 7 and 8, two stable states co-exist (i) high RhoA-GTP, low Rac1-GTP and (ii) low RhoA-GTP, high Rac1-GTP (*Figure 2—figure supplement 2*, panels H and I). Entering area 7 from the BiDR area 3, the system relaxes to the steady state with the higher Rac1-GTP level. Only with the further PAK decrease, a saddle-node bifurcation (see Materials and methods) shifts the system to the alternative steady state with the much lower Rac1-GTP level.

To illustrate the network evolution in response to a gradual decrease in the PAK abundance, we have plotted the trajectories of the Rac1 and RhoA activities, averaged over the cell volume and time (blue curves in *Figure 4B and C*). *Figure 4B* shows that the average Rac1 activity first slowly decreases and then abruptly decays after passing point II (*Figure 4B*). If we follow the Rac1-GTP trajectory in response to increasing PAK inhibitor doses, we obtain a similar curve (*Figure 4—figure supplement 1*). The average RhoA-GTP behaves oppositely, steadily increasing and then jumping to the peak activity after the network passes the BiDR and bistable regions (blue curves in *Figure 4C* and *Figure 4—figure supplement 1*, panel B showing the RhoA-GTP trajectories in response to PAK abundance decrease or IPA-3 increase, respectively). A further decrease in the PAK abundance moves the RhoA-Rac1 network into point III of region 6 with a single steady state of active RhoA and low Rac1 activity (*Figure 4A–C*).

The spatiotemporal dynamic pattern corresponding to point I (*Figure 4B and C*) is a propagating wave illustrated in *Figure 3D–H* and schematically shown in *Figure 4D* where the blue and black arrows illustrate oscillations and the wave propagation along a cell. For point II, the RhoA and Rac1 activity patterns depend on space, but do not change with time (*Figure 4E*). Such spatial dynamics are referred to as a pinning or stalled wave, meaning that a wave of activation first propagates in space, then decelerates and eventually stops, forming stationary RhoA and Rac1 activity profiles (*Mori et al., 2008*) with high steady-state Rac1-GTP at the leading edge (*Figure 4E*). Phenotypically cells maintain a mesenchymal state and polarized shape in both states I and II (*Figure 3I and J*). For point III, the resulting steady-state profile features high RhoA and low Rac1 activities along the entire cell (*Figure 4F*), which is a hallmark of amoeboid cells (*Sahai and Marshall, 2003*; *Sanz-Moreno et al., 2008*; *Wyckoff et al., 2006*). Our results suggest that the transition from the mesenchymal to the amoeboid phenotype becomes switch-like once PAK activity falls below a critical threshold (*Byrne et al., 2016*).

What about the transition back, from point III to point I? Because the underlying GTPase activities show hysteretic behavior, the transition from amoeboid back to the mesenchymal state should follow a different path. Indeed, in our previous study, we observed that a switch from a mesenchymal to amoeboid state occurred at a higher level of PAK inhibition than a switch back when inhibition was gradually reduced (*Byrne et al., 2016*). Our model now can explain the underlying spatiotemporal GTPase dynamics. If cells are forced into the amoeboid state by inhibiting PAK and then allowed to gradually regain PAK activity (red curves in *Figure 4B and C*, and *Figure 4—figure supplement 1*), the network does not pass through the stalled wave state (point II in *Figure 4B and C*). It rather first moves from point III in region 6 through bistable regions 8 and 7, maintaining high RhoA and low Rac1 activities that corresponds to *Figure 4F*. Thus, the network trajectory progresses through alternative states compared to the movement from point I to point III. Upon further relief of PAK inhibition, the network then passes through the BiDR region, and the Rac1 activity jumps to a high value, whereas the RhoA activity switches to a low value, approaching initial point I (*Figure 4B and C*).

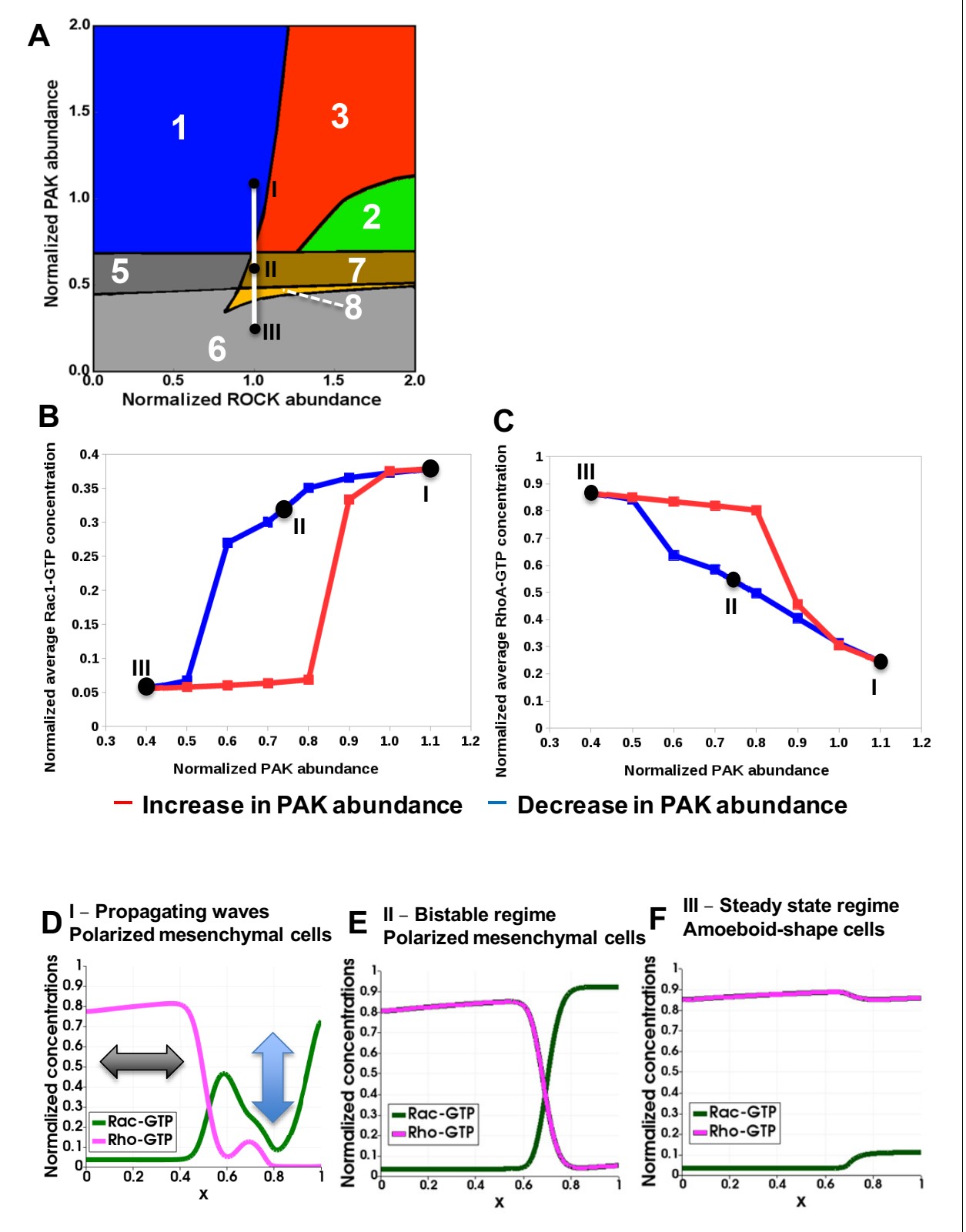

**Figure 4.** Hysteresis of the RhoA and Rac1 activities are manifested upon PAK inhibition and recapitulated by a spatiotemporal model. (**A**) Distinct dynamic regimes of the RhoA-Rac1 network for different DIA and ROCK abundances. Colors and numbers of dynamic regimes are the same as in *Figure 2A*. (**B, C**) Model-predicted dependencies of the RhoA and Rac1 activities on the PAK abundance for gradually decreasing (blue) and increasing (red) PAK abundances. The network evolution occurs through two different routes (blue and red curves in B and C). It is calculated by averaging the

*Figure 4 continued on next page*

*Figure 4 continued*

GTPase activities over the time and cell volume based on western blot data reported in our previous study (**Byrne et al., 2016**). Points I, II and III shown in black (**A**) are also indicated on the network trajectories (**B, C**). (**D–F**) Snapshots of simulated RhoA-GTP and Rac1-GTP spatiotemporal patterns that emerge for different PAK abundances are shown for a 1-D section of a cell. The x axis corresponds to the normalized cell length (**Figure 3A**). Arrows in panel (**D**) illustrate oscillations and the wave propagation along a cell.

The online version of this article includes the following figure supplement(s) for figure 4:

**Figure supplement 1.** Hysteresis of the RhoA and Rac1 activities are manifested upon PAK inhibition.

Summarizing, the experimentally observed hysteresis of RhoA and Rac1 activities upon PAK inhibition is explained by the network evolution through the BiDR and bistable regions. The morphological cell shape changes also follow this pattern. Importantly, bistability in the RhoA-Rac1 network only can be achieved through PAK inhibition, and only when PAK is largely inhibited, cells leave the bistable regions and reach a stable state III where their cell shape becomes amoeboid (**Edelstein-Keshet, 2016**). Our model allows us to systematically dissect the biochemical states that program the GTPase dynamics and resulting cell movement.

## ROCK inhibition results in multiple competing lamellipodia and multipolar cell shapes

Having investigated the consequences of PAK inhibition, we next studied the effects of ROCK inhibition. The model predicts that a decrease in ROCK activity below a certain threshold results in the formation of several oscillatory centers of GTPase activities featuring high (averaged over time) Rac1 activity (**Video 3**). In contrast to periodic RhoGTPase waves propagating from a single Rac1 oscillatory center at the leading edge, several oscillatory Rac1 activity centers result in the uncoordinated and chaotic emergence of waves, thereby preventing a single wave propagation along a cell (compare **Videos 1** and **3**). These findings might imply the emergence of multi-polar cells that extend lamellipodia in several different directions. In fact, multiple competing lamellipodia emerging as a result of ROCK inhibition were previously reported (**Worthylake and Burridge, 2003**).

To determine if ROCK inhibition could induce multiple Rac1-GTP foci, we seeded MDA-MB-231 cells on collagen and treated the cells with the pan-ROCK inhibitor Y-27632. After 15 min, we fixed the cells and stained for active Rac1 and F-actin. Spatially resolved Rac1 activity showed two or three Rac1-GTP poles, whereas cells not incubated with the inhibitor were exclusively mono-polar (**Figure 5A** and **Figure 5—figure supplement 1**, panel A). Using the RhoA-GTP FRET-probe to measure RhoA activity in a spatially resolved manner, **Figure 5B** showed the existence of several centers of uncoordinated RhoA activities. These dynamics are in line with model-predicted patterns (**Video 3**), and in a sharp contrast to cells where ROCK is not inhibited (**Figure 3**, and **Video 1**).

In the absence of ROCK inhibitor, the RhoA-GTP bursts at the cell rear only occur when a propagating wave reaches the rear, that is at low frequency. These bursts cause the cell tail retraction and are associated with the last step of the movement cycle of a polarized elongated cell. When ROCK is inhibited, a GTPase oscillatory center emerges in the tail with the corresponding increase in the frequency of RhoA-GTP bursts (**Figure 5—figure supplement 1**, panels B and C). As a result, a cell loses the ability to retract the tail. These cells do not lose polarity but exhibit substantial morphological changes, acquiring largely elongated shapes (compare **Videos 2** and **4**). In line with these results, our experiments suggest that the total migration distance is smaller for cells treated with ROCK

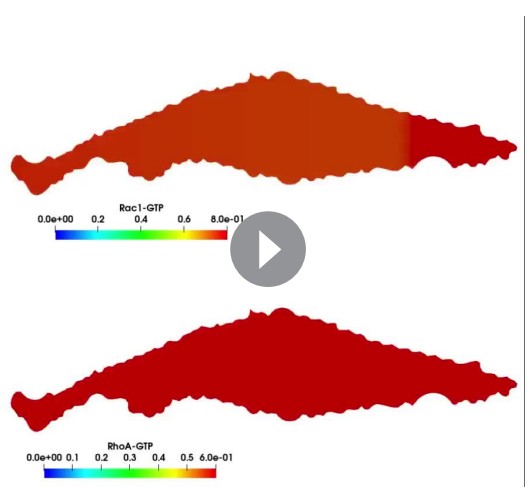

**Video 3.** Model-predicted spatiotemporal activity patterns of RhoA and Rac1 when ROCK is inhibited by 2.5 µM of Y-27632.

https://elifesciences.org/articles/58165#video3

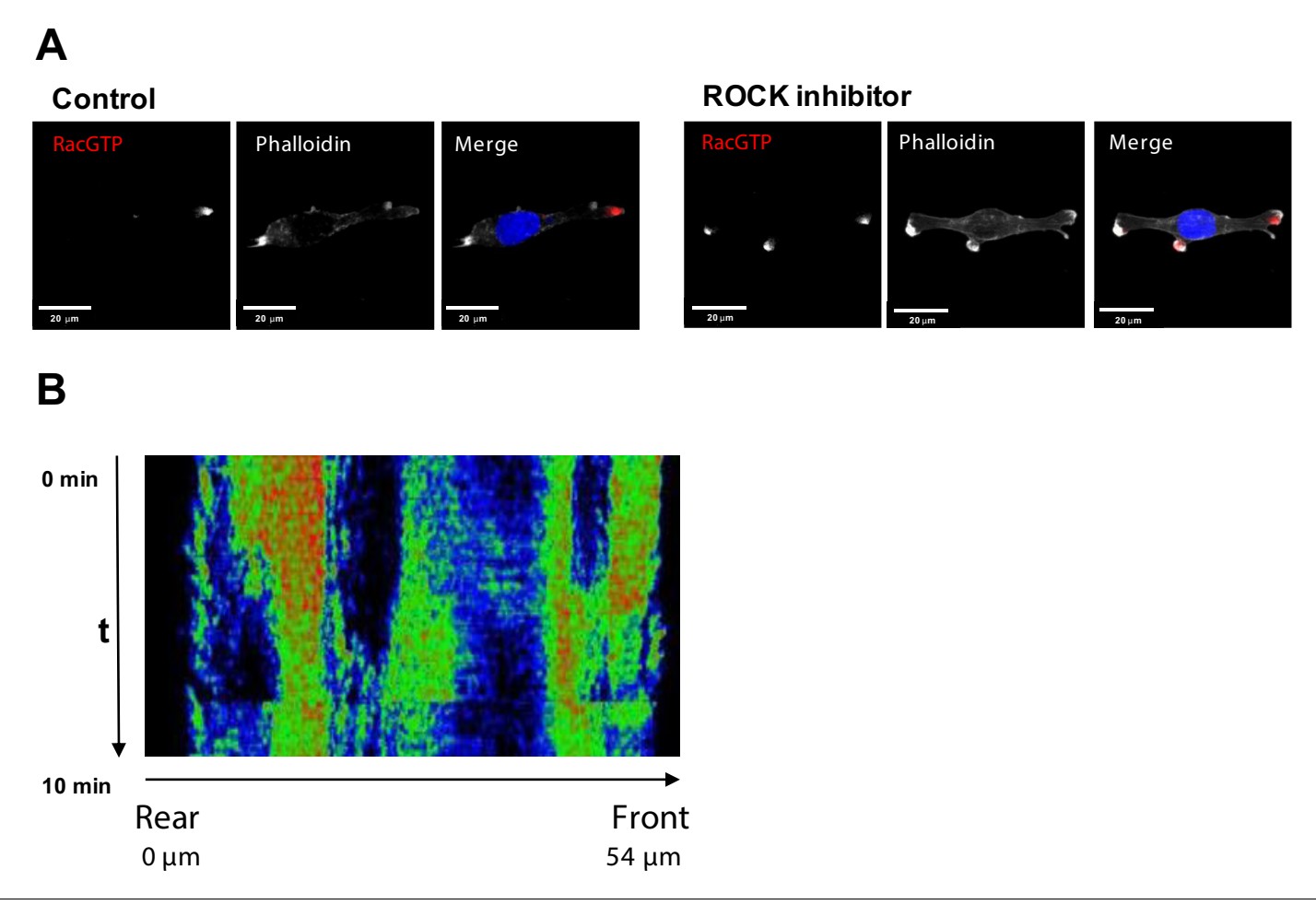

**Figure 5.** Inhibition of ROCK leads to the formation of multi-polar cells. (**A**) Fluorescent microscopy images of Rac1 activity (red), and F-actin (phalloidin, white) and nuclear (DAPI, blue) staining in fixed MDA-MB-213 cells treated or not with 2.5 µM Y-27632 ROCK inhibitor for 15 min. (**B**) Spatiotemporal pattern of the RhoA activity in cells treated with 2.5 µM of ROCK inhibitor Y-27632 measured by the RhoA FRET probe.

The online version of this article includes the following figure supplement(s) for figure 5:

**Figure supplement 1.** Inhibition of ROCK leads to the formation of multi-polar cells.

inhibitor than for untreated cells (*Figure 5—figure supplement 1*, panel D). This decrease can be explained by the formation of multiple lamellipodia and the inability of ROCK-inhibited cells to retract their tail.

Summing up, these data suggest that the ROCK activity above a certain threshold is necessary for the formation of a single high Rac1 activity center at the leading edge and avoiding the appearance of multiple high Rac1 activity centers in a cell. Thus, ROCK cooperates with PAK to maintain the polarized lamellipodia formation and the cell shape typical for mesenchymal cell movement.

## Discussion

RhoGTPases are core regulators of mesenchymal and amoeboid cell migration. They integrate multiple internal and external cues (*Campa et al., 2015*; *Devreotes et al., 2017*; *Lin et al., 2015*; *Park et al., 2017*; *Park et al., 2019*) and relay information to a variety of cellular protein machineries, including proteins driving actin polymerization and cytoskeleton rearrangements, thereby enabling cell migration (*Warner et al., 2019*). Although molecular details of RhoGTPase - effector interactions have been elaborated, we still lack an overall picture of how these GTPase activities and effector interactions are coordinated between the leading and trailing edge in order to enable cell

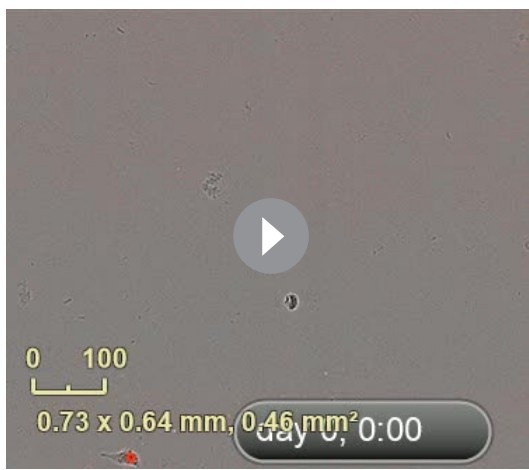

**Video 4.** Live-cell imaging of cellular movement cycles in cells treated with 10 μM Y-27632 ROCK inhibitor. Red color represents staining of the nuclei.
https://elifesciences.org/articles/58165#video4

movement. Here, we present a minimal biochemical mechanism that is necessary and sufficient for the cyclic process of cell migration. This mechanism integrates different temporal dynamics of the RhoA and Rac1 GTPases at the cell front, body and rear and shows how these activities are coordinated by propagating GTPase activation waves. Besides, our model rationalizes how the amoeboid and mesenchymal types of migration interchange by suppression or over-activation of specific RhoA and Rac1 effectors.

A traditional view on mesenchymal migration was that high Rac1 activity persists only at the leading edge, whereas high RhoA activity exists mainly at the rear. This view was supported by the reported mutual antagonism of Rac1 and RhoA (*Byrne et al., 2016*; *Sanz-Moreno et al., 2008*). However, live cell imaging experiments showed oscillations in RhoA activity at the leading edge, challenging the traditional view (*Machacek et al., 2009*; *Pertz, 2010*; *Tkachenko et al., 2011*). Several studies suggested that RhoA not only inhibits Rac1 via ROCK but also activates Rac1 via DIA (*Guilluy et al., 2011a*; *Tsuji et al., 2002*). Our results and literature data (*Brandt et al., 2007*; *Goulimari et al., 2005*; *Newell-Litwa et al., 2015*; *Watanabe et al., 1997*; *Wheeler and Ridley, 2004*) show that the spatial localization of DIA and ROCK is different along the cell; ROCK is more abundant at the cell rear and body, whereas DIA is more abundant at the leading edge that at the rear (*Figure 1C and D*). This difference leads to marked changes in the cellular distribution of RhoA-ROCK versus RhoA-DIA effector complexes (*Figure 1A and B*). Differential localization of DIA and ROCK, as well as different spatial distribution of GEFs, GAPs, and guanosine nucleotide dissociation inhibitors (*de Beco et al., 2018*; *Nikonova et al., 2013*; *Tsyganov et al., 2012*), generate distinct circuitries of RhoA-Rac1 interactions and different RhoA and Rac1 kinetics along a cell (*Figure 2B–F*). Oscillations of RhoA and Rac1 activities at the leading edge guide protrusions and retractions, whereas high, stable RhoA activity and low Rac1-GTP at the rear maintain focal adhesions and the cell attachment to the substrate. Although the distinct RhoGTPase dynamics at the front and rear during a cell migration cycle have been described, it is unknown how exactly a cell integrates these behaviors to coordinate cell movement.

To better understand the kinetic communication between the front and rear, we have developed a model of the RhoGTPase dynamic behaviors in time and space. Our model suggests that periodically repeating RhoGTPase waves connect protrusion-retraction oscillations of RhoA and Rac1 activities at the leading edge and almost stable RhoA and Rac1 activities at the rear. The RhoGTPase waves occur due to diffusion fluxes that are induced by different RhoA-GTP and Rac1-GTP concentrations along the cell and the excitable dynamics of RhoA and Rac1 generated by negative and positive feedback loops in the network (*Tsyganov et al., 2012*). These RhoA and Rac1 activity waves create an autonomous, cyclic mechanism that controls the mesenchymal type of cell migration.

In the initial phase of cell migration, the Rac1-RhoA oscillations push out and retract lamellopodia at the leading edge permitting the cell to explore its environment and follow chemotactic cues (*Machacek et al., 2009*), while high RhoA activity at the trailing edge stabilizes cell adhesion (*Ren et al., 2000*). In the late migration phase, RhoA activity extends toward the front allowing focal adhesions to form at the front, and stress fibers to generate contractile force in the cell body that will retract the rear. At the same time, Rac1 activity traveling toward the trailing edge destabilizes focal adhesions at the rear. The combination of these activities pulls up the rear resulting in cell movement. Their critical coordination is accomplished by the spatially resolved dynamic regulation of the excitable Rac1 and RhoA system described by our mathematical model.

Reaction–diffusion equations have been previously used to describe excitable medium and emerging waves in cellular systems (*FitzHugh, 1961*; *Meinhardt and de Boer, 2001*;

*Nagumo et al., 1962*). In these systems, an activator makes a positive feedback, whereas an inhibitor generates a negative interaction (*Xiong et al., 2010*). Using an activator–inhibitor excitable system, joint waves of cytoskeletal and signaling elements have been modeled (*Bement et al., 2015*; *Graessl et al., 2017*; *Weiner et al., 2007*; *Wu, 2017*). Here, we present a core model of the signaling RhoA – Rac1 system, which captures the formation of RhoA-Rac1 periodic propagating waves that coordinate different signaling dynamics at the cell trailing and leading edges. In our core network, intertwined regulatory connections from RhoGTPase effectors to the GEFs and GAPs can be induced not only by phosphorylation or the formation of protein complexes but can also be mediated by cytoskeletal proteins (*Banerjee and Wedegaertner, 2004*; *Lovelace et al., 2017*; *Mitin et al., 2012*; *Ren et al., 1998*; *Saczko-Brack et al., 2016*). We hypothesize that crosstalk interactions of this core signaling network with cytoskeleton proteins generate actomyosin waves and the cytoskeletal dynamics required for cell migration (*Saha et al., 2018*).

Model predictions are supported by imaging and western blot experiments. Experiments with the RhoA FRET probe corroborated the predictions of RhoA-GTP dynamics at the leading edge (*Figure 3H*) and cell body and rear (*Figure 3I and J* and *Figure 3—figure supplement 1*, panel C). Cell staining with specific Rac1-GTP antibody provided snapshots of Rac1 activity corresponding to protrusion-retraction cycles (*Figure 3K*) and the spreading of Rac1 activity beyond the leading edge into the cell body (*Figure 3L and M* and *Figure 3—figure supplement 1*, panel C, super-resolution microscopy images) as predicted by the model. During a cycle of periodic wave propagation, the model has predicted a greater number of RhoA activity bursts at the cell leading edge than at the cell rear, which is fully supported by our data (*Figure 3J*). Our previous western blot experiments showed the hysteresis of RhoA and Rac1 activities following PAK inhibition and then washing-out the inhibitor (*Byrne et al., 2016*). A reaction-diffusion model of the RhoGTPase dynamics developed here demonstrates the hysteresis of the averaged RhoGTPase activities for the non-stationary spatio-temporal dynamics, – a novel phenomenon previously observed in biology for switches between steady states of bistable systems (*Bagowski and Ferrell, 2001*; *Bhalla et al., 2002*; *Craciun et al., 2006*; *Delbrück, 1949*; *Ferrell, 2002*; *Monod and Jacob, 1961*; *Xiong and Ferrell, 2003*).

Although PAK inhibition (*Figure 4*) induces a transition from the mesenchymal to amoeboid mode of migration and the corresponding changes in the cell shapes (*Byrne et al., 2016*), ROCK inhibition leads to the formation of multiple centers of Rac1 oscillations (*Figure 5*) and multiple competing lamellipodia (*Worthylake and Burridge, 2003*). At the same time, DIA downregulation by siRNA resulted in substantial rewiring of the RhoA-Rac1 signaling network, manifested by an increase in RhoA abundance and a decrease in Rac1 abundance (*Figure 1—figure supplement 1*, panels A and B). Model simulations show that these changes can decrease a threshold DIA abundance required to maintain the initial GTPase dynamics in time and space (*Figure 5—figure supplement 1*, panel E). Thus, cells tend to adapt to DIA1 perturbation by adjusting other protein abundances to keep a minimally perturbed Rho-Rac signaling pattern.

Elegant mathematical models have analyzed the dynamics of small networks of cytoskeleton proteins and GTPases and emerging actin travelling waves (*Barnhart et al., 2017*; *Devreotes et al., 2017*; *Holmes et al., 2012*; *Huang et al., 2013*). It was suggested that dynamics of protrusion-retraction cycles results from coupling of a 'pacemaker' signal transduction and a 'motor' of cytoskeletal networks (*Huang et al., 2013*). These models, together with a more abstract model of generic activators and inhibitors (*Cao et al., 2019*), explained the observed wave-like signal transduction patterns and actin waves, which were localized to the cell front, driving protrusion-retraction cycles (*Miao et al., 2019*). The periodic waves of Rac1-RhoA activities described in this paper propagate through the entire cell, coordinating protrusion-retraction cycles at the front and the adhesion-retraction cycle at the rear, and are different from travelling waves reported previously. These waves also differ from trigger protein phosphorylation waves that propagate in spatially distributed bistable signaling cascades (*Kholodenko, 2009*; *Markevich et al., 2006*; *Muñoz-García et al., 2009*). Another aspect of migration is that cells continuously change shapes during their movement. Wave interactions with deforming cell boundaries will likely modulate the propagation patterns (*Cao et al., 2019*), which can be further analyzed in a future research.

In addition to diffusion and excitable properties of signaling networks, the cell front and rear can communicate via other molecular mechanisms. It was suggested that microtubules can play an important role in the spatial localization of RhoGTPase related proteins and the coordination of front and back signaling (*Cullis et al., 2014*; *Meiri et al., 2012*; *Ren et al., 1998*). Staining intensities of

F-actin, an indicator of Rac signaling, at the front of polarized neutrophils and phosphorylated myosin light chain 2 (pMLC2), an indicator of RhoA signaling, at the cell rear showed that these intensities were neither positively correlated nor anticorrelated (*Wang et al., 2013*). This discovered buffering of the front and rear signaling was completely destroyed by the disruption of microtubules (*Wang et al., 2013*). Different spatial concentration profiles of RhoA and Rac1 downstream effectors considered in our model might depend on the microtubule network.

Mechanical tension and mechano-chemical feedback have also been suggested as mechanisms coordinating behaviors at the cell front and the rear (*Abu Shah and Keren, 2013*; *Bays et al., 2014*; *Collins et al., 2012*; *Guilluy et al., 2011b*; *Lessey et al., 2012*; *Park et al., 2017*; *Saha et al., 2018*; *Warner et al., 2019*). It was proposed that membrane tension is responsible for maintaining front-back polarity rather than diffusible molecules generated at the cell leading edge (*Houk et al., 2012*). However, subsequent work, which exploited a fluid dynamic model with the flow resistance emerging from cytoskeleton-bound transmembrane proteins, showed that membrane tension propagates only locally and fails to mediate long-range signaling (*Shi et al., 2018*). These findings support a view that mechanics only modulates biochemical signaling, as suggested by our model.

In summary, our spatiotemporal model of RhoA-Rac1 signaling proposes how different GTPase dynamics at the cell front and rear are coupled and explains the changes in signaling patterns and cell shapes upon inhibition of GTPase effectors. It represents a minimal, experimentally validated model of the biochemical RhoGTPase network that regulates cell migration. This core biochemical network might be a foundation of detailed mechanistic models that would include many more signaling and cytoskeleton proteins, such as key RhoGEFs and RhoGAPs out of 145 known proteins. A core electro-physiological network model of the heart rhythm and wave propagation (*Noble, 1962*; *Noble, 2007*), as well as a model of cell cycle in *Xenopus* oocytes (*Novak and Tyson, 1993*) captured basic mechanisms of cell's oscillatory machinery and laid the background for more sophisticated and detailed models that now involve tens of ion channels and cell cycle proteins, respectively.

# Materials and methods

**Key resources table**

| Reagent type (species) or resource | Designation | Source or reference | Identifiers | Additional information |
|---|---|---|---|---|
| Antibody | Anti-Rac1 clone 23A8 (Mouse monoclonal) | Millipore | cat.05–389 | (1:500) |
| Antibody | Anti-RhoA (26C4) (Mouse monoclonal) | Santa-Cruz Biotechnology | cat.sc-418 | (1:200) |
| Antibody | anti-GAPDH (D16H11) XP (Rabbit monoclonal) | CST | cat.5174 | (1:3000) |
| Antibody | anti-DIA1 (Rabbit polyclonal) | Thermo | cat.PA5-21409 | WB (1:1500) IF (1:200) |
| Antibody | anti-ROCK1 (Rabbit polyclonal) | Thermo | cat.PA5-22262 | (1:100) |
| Antibody | anti-Rac-GTP (Mouse monoclonal) | New East Bio | cat.26903 | (1:100) |
| Antibody | Anti-mouse F(ab')2 Fragment Alexa Fluor 647 (Goat polyclonal) | Thermo | cat. A-21237 | (1:400) |
| Antibody | Anti-rabbit F(ab')2 Fragment Alexa Fluor 594 (Goat polyclonal) | Thermo | cat. A-11072 | (1:400) |
| Antibody | Anti-rabbit Alexa Fluor-488 (Donkey polyclonal) | Thermo | cat. A-21206 | (1:250) |

*Continued on next page*

Continued

| Reagent type (species) or resource | Designation | Source or reference | Identifiers | Additional information |
|---|---|---|---|---|
| Antibody | Anti-rabbit Alexa Fluor-594 (Goat polyclonal) | Thermo | cat. A-11012 | (1:250) |
| Antibody | Anti-rabbit IgG, HRP-linked (Goat polyclonal) | CST | cat.7074 | (1:10000) |
| Antibody | Anti-mouse IgG, HRP-linked (Horse polyclonal) | CST | cat.7076 | (1:10,000) |
| Strain, strain background (Lentivirus) | IncuCyte NucLight Red Lentivirus Reagent | Essen | Cat. 4625 | |
| Chemical compound, drug | Y-27632 | Sigma Aldrich | Cat.Y0503 | |
| Chemical compound, drug | GSK 269962 | Selleckchem | Cat.S7687 | |
| Chemical compound, drug | 4,6-Diamidino-2-phenylindole dihydrochloride (DAPI), stain | Sigma Aldrich | Cat.10236276001 | (1 µg/ml) |
| Chemical compound, drug | Rhodamine Phalloidin | Thermo | Cat. R415 | |
| Chemical compound, drug | Phalloidin-Alexa Fluor-488 | Thermo | Cat. A12379 | |
| Chemical compound, drug | Puromycin | Sigma Aldrich | Cat. P8833 | |
| Chemical compound, drug | Polibrene | Millipore | Cat.TR-1003-G | |
| Chemical compound, drug | Lipofectamine RNAiMax | Thermo | Cat.13778 | |
| Other | GST-Beads | Sigma Aldrich | G.4510 | |
| Other | Dulbecco's Modified Eagle Medium (DMEM) | Sigma Aldrich | Cat.D6429 | |
| Other | FluoroBrite DMEM Media | Thermo | Cat. A1896701 | |
| Other | Fetal Bovine Serum (FBS) | Gibco | Cat.10270 | |
| Other | Collagen (rat tail) | Sigma Aldrich | Cat.11179179001 | |
| Cell line (Human) | MDA-MB-231 | ATCC | Cat.HTB-26 | Authenticated by the Beatson Institute, Glasgow, UK |
| Transfected Construct (Human) | DIAPH1 siRNA SMART Pool | Dharmacon | cat. L-010347-00-0010 | |
| Recombinant DNA reagent | GST-rhotekin-RBD | Dr. Mike Olson gift (Beatson Institute, Glasgow, UK) | | |
| Recombinant DNA reagent | GST-PAK-CRIB | Dr. Piero Crespo gift (IBBTEC,University of Cantanbria, Spain) | | |
| Recombinant DNA reagent | mTFP-YFP RhoA activity probe | Prof. Olivier Pertz Gift (Institute of Cell Biology, Bern, Switzerland) | | |

*Continued*

| Reagent type (species) or resource | Designation | Source or reference | Identifiers | Additional information |
|---|---|---|---|---|
| Recombinant DNA reagent | psPAX-2 | Prof. Olivier Pertz Gift | | |
| Recombinant DNA reagent | VsVg | Prof. Olivier Pertz Gift | | |
| Software, algorithm | Fiji | *Schindelin et al., 2012* | https://imagej.net/Fiji | |
| Software, algorithm | OpenFOAM | *Weller et al., 1998* | https://www.openfoam.com/ | |
| Software, algorithm | ParaView | *Ayachit, 2015* | https://www.paraview.org/ | |
| Software, algorithm | Salome | *Ribes and Caremoli, 2007* | https://www.salome-platform.org/ | |
| Software, algorithm | Python | | https://www.python.org/ | |
| Software, algorithm | SciPy | *Virtanen et al., 2020* | https://www.scipy.org/ | |
| Software, algorithm | MatplotLib | *Hunter, 2007* | https://matplotlib.org/ | |
| Software, algorithm | OpenCV | *Bradski, 2000* | https://opencv.org/ | |
| Software, algorithm | DYVIPAC | *Nguyen et al., 2015* | https://bitbucket.org/andreadega/dyvipac-python/src/master/ | |
| Software, algorithm | BioNetGen | *Blinov et al., 2004*; *Harris et al., 2016* | https://www.csb.pitt.edu/Faculty/Faeder/?page_id=409 | |

## Experiments

### Tissue Culture and cell treatment

#### Cells

MDA-MB-231 breast cancer cells (a gift from Brad Ozanne, Beatson Institute) were cultured in DMEM supplemented with 2 mM glutamine and 10% fetal calf serum at 37°C in a humidified atmosphere containing 5% $CO_2$. MDA-MB-231 expressing the RhoA activity probe were generated by lentiviral infection of the mTFP-YFP RhoA activity probe (*Fritz et al., 2013*) and selected with puromycin at 2 µg/ml for 3 days. MDA-MB-231 cells with constitutive expression of nuclear mKATE2 were generated by infecting MDA-MB-231 cells with IncuCyte NucLight Red Lentivirus Reagent (Cat. No. 4625) in the presence of polybrene (6 µg/ml, Sigma). After 48 hr, selection was performed by supplement the media with puromycin (2 µg/ml, Sigma). All aforementioned cell lines were mycoplasma negative and tested on a monthly basis when in culture.

#### ROCK inhibition

Cells were incubated with either vehicle, 1 µM GSK 269962 (Tocris) or 2.5, 5 or 10 µM (as indicated in manuscript) Y-27632 (Sigma) for 20 min before the experiments were carried out.

### Knock down by siRNA

Knock-down of DIA1 was achieved by transfecting a smartpool of three siRNAs targeting the human DIAPH1 mRNA and non-targeting siRNA control (Dharmacon cat. L-010347-00-0010). Both siRNAs were transfected at a final concentration of 50 nM using Lipofectamine RNAiMax (Cat.13778) in a 1:2 (v/v) ratio. Cells were kept for 48 hr before the experiments were carried out.

### Rac1 and RhoA pulldowns

MDA-MB-231 and MDA-MB-231 transfected with siRNAs against DIAPH1 were seeded in a 6-well plate coated with rat-tail collagen (see siRNA experiments section) and lysed in 500 µl ice-cold lysis buffer (50 mM Tris-HCl, pH 7.5, 0.2% (v/v) Triton X-100, 150 mM NaCl, 10 mM $MgCl_2$) supplemented with 1 mM protease inhibitors PMSF and leupeptin (Sigma). Cell lysates were cleared of debris by centrifugation for 10 min at 20,000xg at 4°C. 10 µl of the cleared lysate were kept as loading control. The remainder of the lysates were incubated with 6 µl of GST-PAK-CRIB beads for Rac1 pulldowns

or GST-Rhotekin-RBD beads for RhoA pulldowns for 1 hr at 4°C under end-to-end rotation. The GST-PAK-CRIB and GST-Rhotekin-RBD beads were produced as described by *Pellegrin and Mellor, 2008*. The beads were washed with one volume of lysis buffer. The beads and an aliquot of the total lysate as input control were separated by SDS gel electrophoresis using 4–12% NuPAGE precast gels according to the manufacturer's instructions. Gels were electroblotted onto PVDF membranes (Sartorius). Blots were blocked in TBST (50 mM Tris, pH 7.5, 150 mM NaCl, 0.05% Tween-20) containing 5% milk powder and incubated overnight with primary antibody followed by secondary antibodies linked to horse radish peroxidase (HRP). Antibodies used included: Rac1 antibody (Millipore, clone 238A, 1:500), anti-RhoA antibody (Santa Cruz Biotechnology 26C4, sc-418, 1 µg/ml), anti-GAPDH (CST D16H11 XP, diluted 1:3000) and anti-DIA1 (Thermo Fisher cat.PA5-21409, 1 µg/ml). Secondary anti-rabbit and anti-mouse HRP-conjugated antibodies were obtained from CST and used at 1:10,000 dilution. Western Blots were developed using SuperSignal West Femto Maximum Sensitivity Substrate (Thermo Fisher). Images of the blots were acquired in a Bio-Rad ChemiDoc Imager. The western blot bands were quantified using ImageJ.

## Immunofluorescence

Cells were seeded onto high performance glass coverslip, thickness 1 1/2 (Zeiss, cat.474030-9000-000) coated with 0.01% collagen. For ROCK inhibition cells were pretreated as indicated in the corresponding section with Y-27632. Cells were washed twice with PBS, fixed and permeabilized with 3.7% formaldehyde, 0.025% NP-40 in 50 mM Pipes pH6.8, 10 mM $MgCl_2$ for 5 min and blocked in TBS (50 mM Tris, pH 7.5, 150 mM NaCl) containing 2% BSA for 1 hr. Coverslips were incubated overnight in TBS containing 1% BSA with primary anti-Rac1-GTP (New East Bio cat.26903) (1:100), anti-ROCK1 (Thermo cat.PA5-22262) (1:100) or anti-DIA1 (1:200) antibodies. Slides were washed twice with TBS and then incubated for 1 hr at room temperature with secondary antibodies anti-mouse F(ab')2 Fragment Alexa Fluor 647 Conjugate (Thermo cat. A-21237), Donkey anti-rabbit Alexa fluor-488 (Cat. A-21206) or anti-rabbit Alexa fluor-594 (Thermo cat. A-11012) for confocal; anti-rabbit F(ab')2 Fragment Alexa Fluor 594 Conjugate (Thermo cat. A-11072) and anti-mouse F(ab')2 Fragment Alexa Fluor 647 Conjugate for super-resolution microscopy. Slides were washed twice with TBS and incubated with DAPI, 1:100, and phalloidin, conjugated with rhodamine or Alexa Fluor-488 (1:100) (Thermo A12379) for 5 min, washed two times and mounted using VECTASHIELD antifade mounting media (Vector labs Cat. H-1000). Confocal images were taken with an Olympus FV100 or a Nikon A1+ confocal, with 60x oil objective. Super-resolution images were taken with a N-SIM microscope using a with 100x oil objective.

## Proximity ligation assay

The Proximity Ligation Assay (PLA) visualizes an interaction between two proteins that co-localize within $\leq$ 40 nm by an oligonucleotide-mediated ligation and enzymatic amplification reaction whose product is subsequently recognized by a fluorescent probe. Consequently, each fluorescent spot indicates that two proteins are in proximity. The mouse/rabbit Duolink in situ red starter kit (Olink, Uppsala, Sweden) was used according to the manufacturer's instructions. MDA-MB-231 cells were seeded at $1 \times 10^4$ cells per well in a 6-well plate. The cells were fixed and permeabilized as described above for immunofluorescence studies. Then, the cells were incubated with a 1:100 dilution of the primary antibodies (RhoA and DIA) in PBS containing 0.01% BSA overnight at 4°C. For the rest of the protocol the manufacturer's instructions were followed. Briefly, the cells were washed in Buffer A (supplied with the kit) three times for 15 min and incubated with the PLA probes for 1 hr at 37°C in a humidified chamber. This was followed by a 10 min and a 5-min wash in Buffer A. The ligation reaction was carried out at 37°C for 1 hr in a humidified chamber followed by a 10 and 5 min wash in Buffer A. The cells were then incubated with the amplification mix for two hours at 37°C in a darkened humidified chamber. After washing with 1x Buffer B (supplied with the kit) for 10 min and 1 min wash with 0.01x buffer B, followed by 488 phalloidin staining (Molecular Probes Catalog number: A12379) to visualize cellular F-actin, the cells were mounted using the mounting media (containing DAPI to visualize cell nucleus) supplied with the kit. Images were quantified using Fiji distribution of ImageJ. A longitudinal axis emanating at the cell front was drawn through selected cells. Along this axis, the cell was divided into three segments: 10% corresponding to the cell front, 70% corresponding to the cell middle, and 20% corresponding to the cell rear. Then the image was converted

into a 2-bit image and masks over PLA reactions were drawn. Finally, the number of PLA reactions per segment as well as the total area occupied by PLA signals per segment were quantified. All the statistical analyses for PLA were done in Excel.

### Random migration assays

Cell migration assays were performed with cell lines stably expressing nuclear mKATE2 (a red fluorescence protein allowing cell tracking) treated with either vehicle, the ROCK inhibitors Y-27632 (10 µM) or GSK-269962 (1 µM). Cells were seeded on IncuCyte ImageLock 96-well plates (cat.4379) at 100 cells per well and placed into an IncuCyte ZOOM with a dual color filter unit. Images were captured every 10 min using phase contrast and red channel with an 10×/0.25 ph1 objective, over a 24 hr period. Stacks of the red florescence channel were created. ImageJ software was used to enhance contrast, subtract background and transform the images to 8-bit greyscale. Random migration trajectories were obtained from the images using the FastTracks Matlab plugin (*DuChez, 2018*), subsequent statistical analysis and plotting were done in Python.

### Assaying RhoA activity by live-cell FRET imaging

MDA-MB-231 stably expressing the mTFP-YFP RhoA-GTP FRET biosensor (*Kim et al., 2015*) were seeded in Fluorodish glass-bottomed plate (cat.FD35-100) coated with collagen. Cells were treated as indicated for siRNA or ROCK inhibition experiments (Y-27632 2.5 µM). The biosensor-expressing cells were imaged at 5 s intervals for 10 min in an Andor Dragonfly spinning disk confocal microscope with a 60x/1.4 - Oil objective. An excitation wavelength (445 nm) was used for both mTFP and FRET channels, while 480 and 540 nm emission filters were used for the mTFP and FRET channels, respectively, with the Confocal 40 µm High Sensitivity imaging mode. A cell-free area using the same settings for exposure and time was acquired for background correction. The raw images were de-noised with the ImageJ PureDenoise plugin (*Luisier et al., 2010*), and ratiometric images were generated. Kymographs were built using MultiKymographr plugin.

### Modeling

#### Relating the PLA data to the total effector concentrations

The PLA data showed that RhoA interactions with its effectors DIA and ROCK change along the cell from the cell rear to the leading edge (*Figures 1A and B*). This correlates with our experimental data (*Figures 1C and D*) and the literature data on DIA and ROCK localization, suggesting that the concentrations of DIA and ROCK are different at the leading edge, in the middle of the cell, and at the cell rear (*Watanabe et al., 1997*; *Wheeler and Ridley, 2004*; *Brandt et al., 2007*; *Goulimari et al., 2005*; *Newell-Litwa et al., 2015*). The steady-state concentration of the complex of RhoA-GTP ($[Rho\text{-}T]$) and DIA ($[DIA\text{-}Rho\text{-}T]$) can be derived using the rapid equilibrium approximation and the dissociation constant ($K_d^{RhoDIA}$). Taking into account the moiety conservation for DIA, we obtain,

$$[DIA] \cdot [Rho\text{-}T] = K_d^{RhoDIA} \cdot [DIA\text{-}Rho\text{-}T]$$
$$DIA^{tot} = [DIA] + [DIA\text{-}Rho\text{-}T]$$

(1)

Our quantitative proteomic data suggest that the RhoA abundance is at least 10-fold higher than the abundance of all DIA isoforms combined, *Supplementary file 1* (*Byrne et al., 2016*). Therefore, in *Equation 1* we can neglect the changes in the RhoA-GTP concentration caused by the RhoA-GTP sequestration into the complex with DIA. The $K_d^{RhoDIA}$ is at least two orders of magnitude smaller than the RhoA abundance (*Lammers et al., 2008*), which leads to an approximate, linear dependence of the complex concentration on the total DIA abundance

$$[DIA\text{-}Rho\text{-}T] = \frac{DIA^{tot} \cdot [Rho\text{-}T]}{K_d^{RhoDIA} + [Rho\text{-}T]} \sim DIA^{tot}$$

(2)

Thus, our data on the changes in the RhoA-DIA complexes along the cell length at the constant RhoA-GTP level can be interpreted as the changes in the abundance of DIA that can bind RhoA-GTP in the plasma membrane, corroborating the literature data (*Brandt et al., 2007*; *Goulimari et al., 2005*; *Newell-Litwa et al., 2015*; *Watanabe et al., 1997*; *Wheeler and Ridley, 2004*).

The abundance of all ROCK isoforms is also much smaller than the RhoA abundance (see *Supplementary file 1*), which together with the cooperative binding of ROCK domains to active RhoA (*Blumenstein and Ahmadian, 2004*) allows us to conclude that the RhoA-GTP-ROCK complex concentration can also be approximated as a linear function of the total ROCK abundance ($ROCK^{tot}$). Consequently, in the model the total abundances of DIA and ROCK depend on the spatial coordinate along the cell, as shown in *Figures 3B and C*. Associating the x axis with the cell length and considering the y axis along the cell width, we use the following distribution of the DIA and ROCK abundances along the x-axis,

$$DIA^{tot}(x) = (DIA_h - DIA_l) \cdot \frac{x}{L} + DIA_l, \ DIA_h > DIA_l \tag{3}$$

$$ROCK^{tot}(x) = \begin{cases} ROCK_l, \ 0 \leq x \leq x_l \\ ROCK_h, \ x_l \leq x \leq L \end{cases}, \ ROCK_h > ROCK_l$$

where $L$ is the cell length.

## Modeling the RhoA - Rac1 network dynamics

The spatiotemporal dynamics of the RhoA - Rac1 network are governed by a partial differential equation (PDE) system, referred to as a reaction-diffusion model. To derive this PDE system, we first consider ordinary differential equation (ODE) systems that describe biochemical reactions and RhoA and Rac1 interactions with their effectors at any fixed point in the cellular space. The difference between the ODE systems at distinct spatial points is brought about by the changes in the total abundances of ROCK1 and DIA along the longitudinal axis of polarized cells given by *Equation 3* (see also *Figure 3B and C*). These ODE equations are then converted to a PDE system by accounting for the diffusion fluxes of active and inactive protein forms.

The model was populated by the protein abundances from our quantitative mass spectrometry data (*Byrne et al., 2016*). The data suggested that Rac1 and RhoA were the most abundant Rac and Rho isoforms and that their levels exceed the abundances of PAK, ROCK and DIA isoforms combined by an order of magnitude (*Supplementary file 1*). The abundances of ROCK1 and ROCK2 were comparable, DIA1 was the most abundant DIA isoform, and PAK2 was the only detected PAK isoform.

We considered the time scale on which the total abundances of RhoA ($Rho^{tot}$), DIA ($DIA^{tot}$), ROCK ($ROCK^{tot}$), Rac1 ($Rac^{tot}$) and PAK ($PAK^{tot}$) are conserved. We denote active, GTP-bound forms of RhoA and Rac1 by [$Rho$-$T$] and [$Rac$-$T$], and inactive GDP-bound forms by [$Rho$-$D$] and [$Rac$-$D$]. Active forms of DIA, ROCK and active (phosphorylated) PAK are denoted by [$DIA^*$], [$ROCK^*$] and [$pPAK$], respectively. Because of the conservation constraints, the concentrations of active forms can be approximately expressed as the corresponding total abundances minus concentrations of inactive forms. Then, assuming the Michaelis-Menten kinetics for the rates of activation and deactivation reactions of the active forms of the GTPases and their effectors ([$DIA^*$], [$ROCK^*$] and [$pPAK$]), the temporal kinetics of the network are given by the following system of ODEs,

$$\frac{d[Rho\text{-}T]}{dt} = \alpha_{DIA}^{Rho}\alpha_{PAK}^{Rho}V_{GEF}^{Rho}\frac{(Rho^{tot}-[Rho\text{-}T])/K_{GEF}^{Rho}}{1+(Rho^{tot}-[Rho\text{-}T])/K_{GEF}^{Rho}} - V_{GAP}^{Rho}\frac{[Rho\text{-}T]/K_{GAP}^{Rho}}{1+[Rho\text{-}T]/K_{GAP}^{Rho}}$$

$$\frac{d[DIA^*]}{dt} = \alpha_{Rho}^{DIA}V_a^{DIA}\frac{(DIA^{tot}-[DIA^*])/K_a^{DIA}}{1+(DIA^{tot}-[DIA^*])/K_a^{DIA}} - V_i^{DIA}\frac{[DIA^*]/K_i^{DIA}}{1+[DIA^*]/K_i^{DIA}}$$

$$\frac{d[ROCK^*]}{dt} = \alpha_{Rho}^{ROCK}V_a^{ROCK}\frac{(ROCK^{tot}-[ROCK^*])/K_a^{ROCK}}{1+(ROCK^{tot}-[ROCK^*])/K_a^{ROCK}} - V_i^{ROCK}\frac{[ROCK^*]/K_i^{ROCK}}{1+[ROCK^*]/K_i^{ROCK}} \tag{4}$$

$$\frac{d[Rac\text{-}T]}{dt} = \alpha_{DIA}^{Rac}\alpha_{PAK}^{Rac}V_{GEF}^{Rac}\frac{(Rac^{tot}-[Rac\text{-}T])/K_{GEF}^{Rac}}{1+(Rac^{tot}-[Rac\text{-}T])/K_{GEF}^{Rac}} - \alpha_{ROCK}^{Rac}V_{GAP}^{Rac}\frac{[Rac\text{-}T]/K_{GAP}^{Rac}}{1+[Rac\text{-}T]/K_{GAP}^{Rac}}$$

$$\frac{d[pPAK]}{dt} = \alpha_{Rac}^{PAK}V_a^{PAK}\frac{(PAK^{tot}-[pPAK])/K_a^{PAK}}{1+(PAK^{tot}-[pPAK])/K_a^{PAK}} - V_i^{PAK}\frac{[pPAK]/K_i^{PAK}}{1+[pPAK]/K_i^{PAK}}$$

Here, the maximal rates and the Michaelis-Menten constants are denoted by the capital letters V's and K's with relevant indices. These V's values correspond to the maximal rates in the absence of

positive or negative regulatory interactions between GTPases, which modify reaction rates. We describe the regulatory interactions, which specify the negative or positive influence of the active form of protein Y on protein X, by the dimensionless multipliers $\alpha_Y^X$ (illustrated in *Figure 2—figure supplement 1*, panel E) (*Tsyganov et al., 2012*). Assuming general hyperbolic modifier kinetics, each multiplier $\alpha_Y^X$ has the same functional form *Cornish-Bowden, 2012*,

$$\alpha_Y^X = \frac{1 + \gamma_Y^X \cdot Y_a / K_Y^X}{1 + Y_a / K_Y^X} \tag{5}$$

Here, $Y_a$ is active form of protein Y. The coefficient $\gamma_Y^X > 1$ indicates activation; $\gamma_Y^X < 1$ inhibition; and $\gamma_Y^X = 1$ denotes the absence of regulatory interactions, in which case the modifying multiplier $\alpha_Y^X$ equals 1. $K_Y^X$ is the activation or inhibition constant.

## Model-predicted different temporal dynamics of the GTPase activities

Substituting the expressions for modifying multipliers (*Equation 5*) into *Equations 4*, we obtain the following equations governing the temporal dynamics of the active protein forms.

$$
\begin{aligned}
\frac{d[Rho\text{-}T]}{dt} &= V_{GEF}^{Rho} \frac{1 + \gamma_{DIA}^{Rho}[DIA^*]/K_{DIA}^{Rho}}{1 + [DIA^*]/K_{DIA}^{Rho}} \frac{1 + \gamma_{PAK}^{Rho}[pPAK]/K_{PAK}^{Rho}}{1 + [pPAK]/K_{PAK}^{Rho}} \frac{(Rho^{tot} - [Rho\text{-}T])/K_{GEF}^{Rho}}{1 + (Rho^{tot} - [Rho\text{-}T])/K_{GEF}^{Rho}} \\
&\quad - V_{GAP}^{Rho} \frac{[Rho\text{-}T]/K_{GAP}^{Rho}}{1 + [Rho\text{-}T]/K_{GAP}^{Rho}} \\[2mm]
\frac{d[DIA^*]}{dt} &= V_a^{DIA} \frac{1 + \gamma_{Rho}^{DIA}[Rho\text{-}T]/K_{Rho}^{DIA}}{1 + [Rho\text{-}T]/K_{Rho}^{DIA}} \frac{(DIA^{tot} - [DIA^*])/K_a^{DIA}}{1 + (DIA^{tot} - [DIA^*])/K_a^{DIA}} \\
&\quad - V_i^{DIA} \frac{[DIA^*]/K_i^{DIA}}{1 + [DIA^*]/K_i^{DIA}} \\[2mm]
\frac{d[ROCK^*]}{dt} &= V_a^{ROCK} \frac{1 + \gamma_{Rho}^{ROCK}[Rho\text{-}T]/K_{Rho}^{ROCK}}{1 + [Rho\text{-}T]/K_{Rho}^{ROCK}} \frac{(ROCK^{tot} - [ROCK^*])/K_a^{ROCK}}{1 + (ROCK^{tot} - [ROCK^*])/K_a^{ROCK}} \\
&\quad - V_i^{ROCK} \frac{[ROCK^*]/K_i^{ROCK}}{1 + [ROCK^*]/K_i^{ROCK}} \\[2mm]
\frac{d[Rac\text{-}T]}{dt} &= V_{GEF}^{Rac} \frac{1 + \gamma_{DIA}^{Rac}[DIA^*]/K_{DIA}^{Rac}}{1 + [DIA^*]/K_{DIA}^{Rac}} \frac{1 + \gamma_{PAK}^{Rac}[pPAK]/K_{PAK}^{Rac}}{1 + [pPAK]/K_{PAK}^{Rac}} \frac{(Rac^{tot} - [Rac\text{-}T])/K_{GEF}^{Rac}}{1 + (Rac^{tot} - [Rac\text{-}T])/K_{GEF}^{Rac}} \\
&\quad - V_{GAP}^{Rac} \frac{1 + \gamma_{ROCK}^{Rac}[ROCK^*]/K_{ROCK}^{Rac}}{1 + [ROCK^*]/K_{ROCK}^{Rac}} \frac{[Rac\text{-}T]/K_{GAP}^{Rac}}{1 + [Rac\text{-}T]/K_{GAP}^{Rac}} \\[2mm]
\frac{d[pPAK]}{dt} &= V_a^{PAK} \frac{1 + \gamma_{Rac}^{PAK}[Rac\text{-}T]/K_{Rac}^{PAK}}{1 + [Rac\text{-}T]/K_{Rac}^{PAK}} \frac{(PAK^{tot} - [pPAK])/K_a^{PAK}}{1 + (PAK^{tot} - [pPAK])/K_a^{PAK}} \\
&\quad - V_i^{PAK} \frac{[pPAK]/K_i^{PAK}}{1 + [pPAK]/K_i^{PAK}}
\end{aligned}
\tag{6}
$$

Because $DIA^{tot}$ and $ROCK^{tot}$ depend on the spatial coordinate along the cell (*Equation 3*), and $DIA^{tot}$, $ROCK^{tot}$, and $PAK^{tot}$ were perturbed experimentally, we first explored the different possible types of the network temporal dynamics (*Equation 6*) in the parameter space of these three effector abundances. We obtained bifurcation diagrams in each of the three planes of the two effector abundances and classified different types of the dynamic regimes that can be detected (*Figure 2C*, *Figure 4A*, and *Figure 2—figure supplement 1*, panels A-C). We used BioNetGen (*Blinov et al., 2004*; *Harris et al., 2016*) and DYVIPAC (*Nguyen et al., 2015*), software packages, and SciPy (*Oliphant, 2007*) and Matplotlib Python libraries (*Hunter, 2007*). In brief, the sbml file (*Hucka et al., 2018*) describing our ODE model was prepared using BioNetGen software (*Blinov et al., 2004*; *Harris et al., 2016*). Then, the DYVIPAC python software package (*Nguyen et al., 2015*) was used to sample a 2-D parameter space and to determine the number and the stability types of steady states for each sample point in this parameter space. The DYVIPAC algorithm allowed detecting only local bifurcations (*Kuznetsov, 2004*), and the obtained sampling data served as an input to a python script, which plotted initial two-paramter bifurcation diagrams. To reveal the borders of non-local bifurcations, for example saddle homoclinic bifurcation (*Nekorkin, 2015*), we analyzed the phase portraits of the system by plotting nullclines, vector fields and limit cycles generated using python scripts. Then, the necessary changes to the bifurcation diagrams were done manually to

include borders for non-local bifurcations. The code that performs calculations is provided in the Supplemental Information.

To get initial insights into different dynamic regimes of this 5 ODE system (*Equation 6*), we analyzed the vector fields and the nullclines for a 2 ODE system, obtained using the quasi steady-state approximation. Because the concentrations of active forms of DIA, ROCK and PAK are an order of magnitude less than the GTPase concentrations, this allows us to introduce a small parameter into our ODE system. Using the Tikhonov theorem (*Tikhonov, 1952*; *Tikhonov et al., 1985*), we can express these active effector concentrations in terms of $[Rho-T]$ and $[Rac-T]$ by applying the quasi steady-state approximation, as follows (*Tsyganov et al., 2012*),

$$\begin{cases} \frac{d[DIA^*]}{dt} = 0 \\ \frac{d[ROCK^*]}{dt} = 0 \\ \frac{d[pPAK]}{dt} = 0 \end{cases} \rightarrow \begin{cases} [DIA^*] = f_{DIA}([Rho\text{-}T], \ [Rac\text{-}T]) \\ [ROCK^*] = f_{ROCK}([Rho\text{-}T], \ [Rac\text{-}T]) \\ [pPAK] = f_{PAK}([Rho\text{-}T], \ [Rac\text{-}T]) \end{cases} \tag{7}$$

To find the functions, $f_{DIA}$, $f_{ROCK}$, and $f_{PAK}$, *Equation 7* were solved numerically for each value of active RhoA and Rac1. The solutions were substituted into the equations governing the dynamics of RhoA-GTP and Rac1-GTP (see *Equation 6*) to obtain the following system of only two differential equations.

$$\begin{aligned} \frac{d[Rho\text{-}T]}{dt} &= V_{GEF}^{Rho} \frac{1 + \gamma_{DIA}^{Rho} f_{DIA}/K_{DIA}^{Rho}}{1 + f_{DIA}/K_{DIA}^{Rho}} \frac{1 + \gamma_{PAK}^{Rho} f_{PAK}/K_{PAK}^{Rho}}{1 + f_{PAK}/K_{PAK}^{Rho}} \frac{(Rho^{tot} - [Rho\text{-}T])/K_{GEF}^{Rho}}{1 + (Rho^{tot} - [Rho\text{-}T])/K_{GEF}^{Rho}} \\ &\quad - V_{GAP}^{Rho} \frac{[Rho\text{-}T]/K_{GAP}^{Rho}}{1 + [Rho\text{-}T]/K_{GAP}^{Rho}} \\ \frac{d[Rac\text{-}T]}{dt} &= V_{GEF}^{Rac} \frac{1 + \gamma_{DIA}^{Rac} f_{DIA}/K_{DIA}^{Rac}}{1 + f_{DIA}/K_{DIA}^{Rac}} \frac{1 + \gamma_{PAK}^{Rac} f_{PAK}/K_{PAK}^{Rac}}{1 + f_{PAK}/K_{PAK}^{Rac}} \frac{(Rac^{tot} - [Rac\text{-}T])/K_{GEF}^{Rac}}{1 + (Rac^{tot} - [Rac\text{-}T])/K_{GEF}^{Rac}} \\ &\quad - V_{GAP}^{Rac} \frac{1 + \gamma_{ROCK}^{Rac} f_{ROCK}/K_{ROCK}^{Rac}}{1 + f_{ROCK}/K_{ROCK}^{Rac}} \frac{[Rac\text{-}T]/K_{GAP}^{Rac}}{1 + [Rac\text{-}T]/K_{GAP}^{Rac}} \end{aligned} \tag{8}$$

*Figure 2—figure supplement 2*, panels A-I illustrate the vector fields and nullclines for a 2-D system describing the temporal dynamics of RhoA-GTP and Rac1-GTP. Each dynamic regime shown in *Figures 2C* and *4A* and *Figure 2—figure supplement 1*, panels A-C has the corresponding phase portrait in *Figure 2—figure supplement 2*. The red line represents the solution for the equation $d[Rho-T]/dt = 0$ (the RhoA nullcline), and the blue line represents the solution for the equation $d[Rac-T]/dt = 0$ (the Rac1 nullcline).

Points of intersection of the nullclines are network steady states for both 5 ODE and 2 ODE systems. These states can be stable or unstable (shown by bold points or triangles, respectively in *Figure 2—figure supplement 2*, panels A-I). For each of dynamic regimes 0, 1 and 6 there is only a single steady state, which is a stable focus for regime 0, stable node for regime 6 and an unstable focus for regime 1 (points 1 at *Figure 2—figure supplement 2*, panels A, B and G). If a steady state is unstable focus, self-sustained oscillations (a limit cycle) may or may not exist in the system, depending on the global topology of the vector fields. In our system, although unstable focus steady states are observed in regimes 1–5 and 7, self-sustained oscillations exist only in regimes 1 and 3. For these oscillatory regimes, we plotted projections of the limit cycle trajectory calculated for a five-dimensional ODE system (*Equation 6*) to a two-dimensional space of active RhoA and active Rac1 concentrations (green curves in *Figure 2—figure supplement 2*, panels B and D). 1-D bifurcation diagrams presented in *Figure 2—figure supplement 3* illustrate transitions between these different regimes.

The increase in the DIA abundance at low, fixed ROCK abundance can transform dynamic regime 0 into dynamic regime 1 (*Figure 2—figure supplement 1*, panel A and *Figure 2—figure supplement 3*, panels C and D), following the Andronov-Hopf bifurcation (*Kuznetsov, 2004*). This bifurcation results in losing the stability of the focus (point 1, *Figure 2—figure supplement 2*, panels A and B) and the appearance of a stable limit cycle around the unstable focus (green trajectory, *Figure 2—figure supplement 2*, panel B, dashed lines, *Figure 2—figure supplement 3*, panels C and D). The point with the coordinates (1, 1) in *Figure 2A* and *Figure 2—figure supplement 1*, panel A is the 'physiological point' of sustained oscillations at the leading edge.

An increase in the ROCK abundance at fixed DIA abundance can transform dynamic regime 1 into regime 3 termed BiDR (*Figure 2C* and *Figure 2—figure supplement 3*, panels A and B). At certain enlarged ROCK abundances, the Rac1 nullcline crosses the RhoA nullcline generating a saddle point and a stable node (points 2 and 3, *Figure 2—figure supplement 2*, panel D), known as a saddle-node bifurcation (*Kuznetsov, 2004*). In the BiDR regime a stable limit cycle coexists with a stable node, and each of these dynamic regimes has its own basin of attraction (*Figure 2—figure supplement 2*, panels D, and *Figure 2—figure supplement 3*, panels A and B). A saddle point separates the basins of attraction of the limit cycle and the stable node. The further increase in the ROCK abundance moves the system to regime 2 (*Figure 2—figure supplement 3*, panels A and B) where the limit cycle disappears, whereas an unstable focus (point 1, *Figure 2—figure supplement 2*, panel C), saddle (point 2, *Figure 2—figure supplement 2*, panel C) and stable node (point 3, *Figure 2—figure supplement 2*, panel C) persist. The disappearance of the limit cycle occurs when it merges with a saddle point in the process termed as a saddle homoclinic bifurcation (*Nekorkin, 2015*). Thus, although regimes 2 and 3 have the same number and stability types of the steady-state solutions, a stable limit cycle exists only in regime 3. Whereas in regime 2, there is no stable limit cycle, perturbations to Rac1 can result in generation of overshooting Rac1 activity pulses before the actviity returns to the stable steady state. These pulses occur when the system trajectory follows the vector-field around the unstable focus (point 1, *Figure 2—figure supplement 2*, panel C). Thus, although regime 2 is monostable, it creates excitable media that supports the propagation of excitable activity pulses.

If the DIA abundance increases at the high, fixed ROCK abundance, a saddle-node bifurcation appears earlier than the Andronov-Hopf bifurcation, and dynamic regime 1 with single stable focus transforms into dynamic regime 4 (*Figure 2—figure supplement 1*, panel A, and *Figure 2—figure supplement 3*, panels E and F) with stable node (point 3, *Figure 2—figure supplement 2*, panel E) and saddle point (point 2, *Figure 2—figure supplement 2*, panel E) in addition to the stable focus (point 1, *Figure 2—figure supplement 2*, panel E). At some point (*Figure 2—figure supplement 1*, panel B) dynamic regimes 0–4 converge, the saddle-node, the saddle homoclinic and the Andronov-Hopf bifurcations happen simultaneously in a process known as the Bogdanov-Takens bifurcation (*Kuznetsov, 2004*).

Regimes 4 and 8 have two stable steady states (points 1 and 3, *Figure 2—figure supplement 2*, panels E and I) and one saddle point (point 2, *Figure 2—figure supplement 2*, panels E and I), which separates the basins of attraction of the stable states. Regime 8 is a classic bistability regime arising from a double negative feedback in the RhoA-Rac1 network (*Figure 2—figure supplement 3*, panels E and F). One stable node has the high RhoA and low Rac activities, whereas the other stable node has the high Rac and low Rho activities (points 1 and 3, *Figure 2—figure supplement 2*, panel I) (*Byrne et al., 2016*). In regime 4, one of the stable steady states is a stable node, whereas the other is a stable focus. Both stable states have low Rac1-GTP levels, but the stable focus (point 1, *Figure 2—figure supplement 2*, panel E) has a low RhoA-GTP level, while the stable node (point 3, *Figure 2—figure supplement 2*, panel E) has a high RhoA-GTP level. Regime 4 occurs for low DIA abundances, when the activating connection from RhoA to Rac1 is weak. The dynamical behavior of regime 7 is similar to the dynamics of regime 8. Both regimes exhibit two stable nodes (points 3 and 5 for regime 7, *Figure 2—figure supplement 2*, panel H) and a saddle resulting in bistability. Regime 7 has an additional unstable focus and saddle (points 1–2, *Figure 2—figure supplement 2*, panel H), which do not substantially change the basins of attraction of stable nodes.

Regime 6 has a single steady state that is a stable node, to which all solutions converge regardless of the initial conditions (*Figure 2—figure supplement 2*, panel G). The dynamical behavior of regime 5 is similar to the dynamics of regime 6. Regime 5 has a single stable node but also an additional unstable focus and saddle (points 1–2, *Figure 2—figure supplement 2*, panel F), which does not substantially change the basin of attraction of the stable node.

Summarizing, the above analysis of a 2-D system (*Equation 8*) helped us comprehend the dynamic behaviors and parameter bifurcation diagrams obtained for a 5-D system (*Equation 6*, in *Figures 2C* and *4A* and *Figure 2—figure supplement 1*, panels A-C).

## Describing spatiotemporal dynamical regimes in the model

To explore the spatiotemporal behavior of the RhoA-Rac1 network in an entire cell, we took into account diffusion fluxes and spatial distribution of RhoA, Rac1 and their effectors. The

spatiotemporal dynamics of the system is described by the following system of partial differential equations (PDEs). Since active and inactive forms of RhoA and Rac1 GTPases can have different diffusion coefficients, the PDEs include both protein forms.

$$
\frac{\partial [Rho\text{-}T]}{\partial t} = V_{GEF}^{Rho} \frac{1 + \gamma_{DIA}^{Rho}[DIA^*]/K_{DIA}^{Rho}}{1 + [DIA^*]/K_{DIA}^{Rho}} \frac{1 + \gamma_{PAK}^{Rho}[pPAK]/K_{PAK}^{Rho}}{1 + [pPAK]/K_{PAK}^{Rho}} \frac{[Rho\text{-}D]/K_{GEF}^{Rho}}{1 + [Rho\text{-}D]/K_{GEF}^{Rho}}
$$

$$
- V_{GAP}^{Rho} \frac{[Rho\text{-}T]/K_{GAP}^{Rho}}{1 + [Rho\text{-}text\text{-}T]/K_{GAP}^{Rho}} - \nabla(-D_{RhoT}\nabla[Rho\text{-}T])
$$

$$
\frac{\partial [Rho\text{-}D]}{\partial t} = -V_{GEF}^{Rho} \frac{1 + \gamma_{DIA}^{Rho}[DIA^*]/K_{DIA}^{Rho}}{1 + [DIA^*]/K_{DIA}^{Rho}} \frac{1 + \gamma_{PAK}^{Rho}[pPAK]/K_{PAK}^{Rho}}{1 + [pPAK]/K_{PAK}^{Rho}} \frac{[Rho\text{-}D]/K_{GEF}^{Rho}}{1 + [Rho\text{-}D]/K_{GEF}^{Rho}}
$$

$$
+ V_{GAP}^{Rho} \frac{[Rho\text{-}T]/K_{GAP}^{Rho}}{1 + [Rho\text{-}T]/K_{GAP}^{Rho}} - \nabla(-D_{RhoD}\nabla[Rho\text{-}D])
$$

$$
\frac{\partial [DIA^*]}{\partial t} = V_a^{DIA} \frac{1 + \gamma_{Rho}^{DIA}[Rho\text{-}T]/K_{Rho}^{DIA}}{1 + [Rho\text{-}T]/K_{Rho}^{DIA}} \frac{(DIA^{tot}(\vec{x}) - [DIA^*])/K_a^{DIA}}{1 + (DIA^{tot}(\vec{x}) - [DIA^*])/K_a^{DIA}}
$$

$$
- V_i^{DIA} \frac{[DIA^*]/K_i^{DIA}}{1 + [DIA^*]/K_i^{DIA}}
$$

$$
\frac{\partial [ROCK^*]}{\partial t} = V_a^{ROCK} \frac{1 + \gamma_{Rho}^{ROCK}[Rho\text{-}T]/K_{Rho}^{ROCK}}{1 + [Rho\text{-}T]/K_{Rho}^{Rock}} \frac{(ROCK^{tot}(\vec{x}) - [ROCK^*])/K_a^{ROCK}}{1 + (ROCK^{tot}(\vec{x}) - [ROCK^*])/K_a^{ROCK}} \tag{9}
$$

$$
- V_i^{ROCK} \frac{[ROCK^*]/K_i^{ROCK}}{1 + [ROCK^*]/K_i^{ROCK}}
$$

$$
\frac{\partial [Rac\text{-}T]}{\partial T} = V_{GEF}^{Rac} \frac{1 + \gamma_{DIA}^{Rac}[DIA^*]/K_{DIA}^{Rac}}{1 + [DIA^*]/K_{DIA}^{Rac}} \frac{1 + \gamma_{PAK}^{Rac}[pPAK]/K_{PAK}^{Rac}}{1 + [pPAK]/K_{PAK}^{Rac}} \frac{[Rac\text{-}D]/K_{GEF}^{Rac}}{1 + [Rac\text{-}D]/K_{GEF}^{Rac}}
$$

$$
- V_{GAP}^{Rac} \frac{1 + \gamma_{ROCK}^{Rac}[ROCK^*]/K_{ROCK}^{Rac}}{1 + [ROCK^*]/K_{ROCK}^{Rac}} \frac{[Rac\text{-}T]/K_{GAP}^{Rac}}{1 + [Rac\text{-}T]/K_{GAP}^{Rac}} - \nabla(-D_{RacT}\nabla[Rac\text{-}T])
$$

$$
\frac{\partial [Rac\text{-}D]}{\partial T} = -V_{GEF}^{Rac} \frac{1 + \gamma_{DIA}^{Rac}[DIA^*]/K_{DIA}^{Rac}}{1 + [DIA^*]/K_{DIA}^{Rac}} \frac{1 + \gamma_{PAK}^{Rac}[pPAK]/K_{PAK}^{Rac}}{1 + [pPAK]/K_{PAK}^{Rac}} \frac{[Rac\text{-}D]/K_{GEF}^{Rac}}{1 + [Rac\text{-}D]/K_{GEF}^{Rac}}
$$

$$
+ V_{GAP}^{Rac} \frac{1 + \gamma_{ROCK}^{Rac}[ROCK^*]/K_{ROCK}^{Rac}}{1 + [ROCK^*]/K_{ROCK}^{Rac}} \frac{[Rac\text{-}T]/K_{GAP}^{Rac}}{1 + [Rac\text{-}T]/K_{GAP}^{Rac}} - \nabla(-D_{RacD}\nabla[Rac\text{-}D])
$$

$$
\frac{\partial [pPAK]}{\partial t} = V_a^{PAK} \frac{1 + \gamma_{Rac}^{PAK}[Rac\text{-}T]/K_{Rac}^{PAK}}{1 + [Rac\text{-}T]/K_{Rac}^{PAK}} \frac{(PAK^{tot} - [pPAK])/K_a^{PAK}}{1 + (PAK^{tot} - [pPAK])/K_a^{PAK}}
$$

$$
- V_i^{PAK} \frac{[pPAK]/K_i^{PAK}}{1 + [pPAK]/K_i^{PAK}}
$$

Here, $D_{RhoT}$ and $D_{RhoD}$ are the diffusion coefficients of active and inactive forms of RhoA, and $D_{RacT}$ and $D_{RacD}$ are the diffusion coefficients of active and inactive forms of Rac1. For all forms of RhoA and Rac1, zero-gradient boundary conditions are considered at the boundaries of the computational domain, describing no flux conditions at the cell borders. The spatial profiles of the total DIA and ROCK concentrations are set by *Equation 3* that describes their distribution along the cell length (the x-axis). No other total abundances vary with the cell length in our reaction-diffusion model.

At the leading edge, the total concentrations of DIA and ROCK correspond to oscillatory regimes 1 and 3 observed for a well-mixed system (*Figure 2A, B and F*, and *Figure 2—figure supplement 2*, panels B and D). In the spatial case, the PDE equations (*Equation 9*) with these parameters generate excitable media, where self-sustained waves of the RhoA and Rac1 activities are formed periodically. Thus, the leading edge can be considered as a 'pacemaker' of the GTPase cellular machinery (*Huang et al., 2013*), by analogy to the sinoatrial node in the heart (*Mangoni and Nargeot, 2008*).

At the cell body and rear the total concentration of DIA is lower, and the total concentration of ROCK is higher than at the leading edge. For the well-mixed system (*Equation 6*), these concentration parameters correspond to regime 2 (*Figure 2A and C* and *Figure 2—figure supplement 2*, panel C). For the dynamics in space and time, these parameters bring about weakly excitable media, which can propagate self-sustained waves of RhoA and Rac1 activities after receiving a strong stimulus, but unable to autonomously generate such waves. In the stimulus absence, high RhoA and low Rac1 stationary activities are maintained in this media. Following an over-threshold stimulus, this weakly excitable media propagates the wave of high Rac1 activity, and then returns to the steady state with high RhoA and low Rac1 activities. Importantly, the excitability of this media gradually

decays approaching the cell rear. As a result, in a mesenchymal polarized cell a number of waves of RhoA and Rac1 activity must be generated at the leading edge to induce a self-sustained wave in the cell body and rear, in contrast with the heart where every wave generated in the sinoatrial node spreads through the entire heart. The higher concentration of ROCK exists at the cell body and rear, the higher number of waves must be generated at the leading edge before a GTPase activity wave propagates through an entire cell. If the total ROCK concentration of is too high in the cell body and rear, the waves generated at the leading edge vanish before propagating deeply into the cell and reaching the cell rear.

Thus, high excitability at the leading edge and low excitability in the cell body and at the rear result in a cyclic dynamic pattern, in which multiple protrusion-retraction cycles are generated at the leading edge before a migrating cell moves.

## Modeling the mechanisms of PAK and ROCK inhibition

The mechanism of PAK inhibition by allosteric inhibitor IPA-3 was modeled similarly as in our previous study (*Byrne et al., 2016*). IPA-3 reversibly binds to an inactive PAK conformation, and prevents PAK activation (*Deacon et al., 2008*; *Viaud and Peterson, 2009*). Assuming rapid equilibrium of inactive PAK – inhibitor complex, the effect of PAK inhibitor IPA-3 is modelled by considering the concentration of inactive PAK as the following function of [IPA-3],

$$[PAK]([IPA-3]) = \frac{PAK|_{[IPA-3]=0}}{\left(1 + \frac{[IPA-3]}{K_I^{PAK}}\right)} \tag{10}$$

Both, ATP competitive ROCK inhibitor Y-27632 and ATP bind to an active conformation of the ROCK kinase (*Yamaguchi et al., 2006*; *Ishizaki et al., 2000*). Thus when Y-27632 is present, the decrease in the ROCK kinase activity can be described by the following multiplier, $\beta<1$,

$$\beta = \left(1 + \frac{[ATP]}{K_d^{ATP}}\right) \Big/ \left(1 + \frac{[ATP]}{K_d^{ATP}} + \frac{[Y-27632]}{K_I^{ROCK}}\right) \tag{11}$$

## Dimensionless equations

To reduce the number of parameters, we express the PDE system, *Equation 9*, in a dimensionless form, *Equation 10* (*Barenblatt, 2003*). To simplify the interpretation of numerical results, we left the time as the only dimensional variable (measured in seconds) that directly corresponds to the time, measured in experiments.

$$\frac{\partial rho}{\partial t} = v_{GEF}^{Rho} \frac{1 + \gamma_{DIA}^{Rho} dia/k_{DIA}^{Rho}}{1 + dia/k_{DIA}^{Rho}} \frac{1 + \gamma_{PAK}^{Rho} pak/k_{PAK}^{Rho}}{1 + pak/k_{PAK}^{Rho}} \frac{rhod/k_{GEF}^{Rho}}{1 + rhod/k_{GEF}^{Rho}} - v_{GAP}^{Rho} \frac{rho/k_{GAP}^{Rho}}{1 + rho/k_{GAP}^{Rho}}$$
$$- \nabla(-d_{Rho}\nabla_{rho})$$

$$\frac{\partial rhod}{\partial t} = -v_{GEF}^{Rho} \frac{1 + \gamma_{DIA}^{Rho} dia/k_{DIA}^{Rho}}{1 + dia/k_{DIA}^{Rho}} \frac{1 + \gamma_{PAK}^{Rho} pak/k_{PAK}^{Rho}}{1 + pak/k_{PAK}^{Rho}} \frac{rhod/k_{GEF}^{Rho}}{1 + rhod/k_{GEF}^{Rho}} + v_{GAP}^{Rho} \frac{rho/k_{GAP}^{Rho}}{1 + rho/k_{GAP}^{Rho}}$$
$$- \nabla(-d_{RhoD}\nabla_{rhoD})$$

$$\frac{\partial dia}{\partial t} = v_a^{DIA} \frac{1 + \gamma_{Rho}^{DIA} rho/k_{Rho}^{DIA}}{1 + rho/k_{Rho}^{DIA}} \frac{(d(\vec{X}) - dia)/k_a^{DIA}}{1 + (d(\vec{X}) - dia)/k_a^{DIA}} - v_i^{DIA} \frac{dia/k_i^{DIA}}{1 + dia/k_i^{DIA}}$$

$$\frac{\partial rock}{\partial t} = v_a^{Rock} \frac{1 + \gamma_{Rho}^{ROCK} rho/k_{Rho}^{ROCK}}{1 + rho/k_{Rho}^{ROCK}} \frac{(r(\vec{X}) - rock)/k_a^{ROCK}}{1 + (r(\vec{X}) - rock)/k_a^{ROCK}} - v_i^{ROCK} \frac{rock/k_i^{ROCK}}{1 + rock/k_i^{ROCK}}$$

$$\frac{\partial rac}{\partial t} = v_{GEF}^{Rac} \frac{1 + \gamma_{DIA}^{Rac} dia/k_{DIA}^{Rac}}{1 + dia/k_{DIA}^{Rac}} \frac{1 + \gamma_{PAK}^{Rac} pak/k_{PAK}^{Rac}}{1 + pak/k_{PAK}^{Rac}} \frac{racd/k_{GEF}^{Rac}}{1 + racd/k_{GEF}^{Rac}}$$
$$- v_{GAP}^{Rac} \frac{1 + \gamma_{ROCK}^{Rac} \beta rock/k_{ROCK}^{Rac}}{1 + \beta rock/k_{ROCK}^{Rac}} \frac{rac/k_{GAP}^{Rac}}{1 + rac/k_{GAP}^{Rac}} - \nabla(-d_{Rac}\nabla_{rac})$$

$$\frac{\partial racd}{\partial t} = -v_{GEF}^{Rac} \frac{1 + \gamma_{DIA}^{Rac} dia/k_{DIA}^{Rac}}{1 + dia/k_{DIA}^{Rac}} \frac{1 + \gamma_{PAK}^{Rac} pak/k_{PAK}^{Rac}}{1 + pak/k_{PAK}^{Rac}} \frac{racd/k_{GEF}^{Rac}}{1 + racd/k_{GEF}^{Rac}}$$
$$+ v_{GAP}^{Rac} \frac{1 + \gamma_{ROCK}^{Rac} \beta rock/k_{ROCK}^{Rac}}{1 + \beta rock/k_{ROCK}^{Rac}} \frac{rac/k_{GAP}^{Rac}}{1 + rac/k_{GAP}^{Rac}} - \nabla(-d_{RacD}\nabla_{racd})$$

(12)

$$\frac{\partial pak}{\partial t} = v_a^{PAK} \frac{1 + \gamma_{Rac}^{PAK} rac/k_{Rac}^{PAK}}{1 + rac/k_{Rac}^{PAK}} \frac{(p-pak)/(K_a^{PAK}(1+I_{PAK}))}{1 + (p-pak)/(K_a^{PAK}(1+I_{PAK}))} - v_i^{PAK} \frac{pak/k_i^{PAK}}{1 + pak/k_i^{PAK}}$$

$$rho = \frac{[Rho\text{-}T]}{Rho^{tot}}, rhod = \frac{[Rho\text{-}D]}{Rho^{tot}}, rac = \frac{[Rac\text{-}T]}{Rac^{tot}}, racd = \frac{[Rac\text{-}D]}{Rac^{tot}}, pak = \frac{[pPAK]}{PAK^{tot}}$$

$$dia = \frac{[DIA^*]}{DIA^{tot}}, rock = \frac{[ROCK^*]}{ROCK^{tot}}, p = \frac{PAK^{tot}}{PAK^{total}}, \vec{X} = \vec{x}/L, I_{PAK} = \frac{[IPA\text{-}3]}{K_I^{PAK}}$$

$$\beta = \left(1 + \frac{[ATP]}{K_d^{ATP}}\right) / \left(1 + \frac{[ATP]}{K_d^{ATP}} + I_{ROCK}\right), I_{ROCK} = \frac{[Y\text{-}27632]}{K_I^{ROCK}}$$

$$d_{Rho} = \frac{D_{RhoT}}{L^2}, d_{RhoD} = \frac{D_{RhoD}}{L^2}, d_{Rac} = \frac{D_{RacT}}{L^2}, d_{RacD} = \frac{D_{RacD}}{L^2}$$

$$d(x) = (d_h - d_l) \cdot X + d_l, \ r = \begin{cases} r_l, \ 0 \le X \le X_l \\ r_h, \ X_l \le X \le 1 \end{cases}$$

$$v_Y^X = V_Y^X/X^{tot}, \ X = Rho, DIA, ROCK, Rac, PAK, \ Y = GEF, GAP, a, i$$

$$k_Y^X = K_Y^X/X^{tot}, \ X = Rho, DIA, ROCK, Rac, PAK$$

$$Y = Rho, DIA, ROCK, Rac, PAK, GEF, GAP, i, a$$

The parameters are listed in *Supplementary file 2*. Many parameters were taken from our previous mechanistic model of the RhoA-Rac1 network, which was tailored to MDA-MB-231 cells (*Byrne et al., 2016*), – the same cell line we used in this work. As in the previous model, we used quantitative mass spectrometry data to determine the prevailing protein isoforms of the RhoA-Rac1 network in MDA-MB-231 cells, as well as the protein abundances. The parameters of activation and deactivation of RhoA, Rac1 and PAK were estimated based on the literature data (*Lyda et al., 2019*; *Tang et al., 2018*). The parameters of activation and deactivation of DIA and ROCK were estimated based on typical association and dissociation constants of protein-protein interactions (*Kholodenko et al., 1999*). The parameters of hyperbolic multipliers (*Equation 5*) were estimated based on the parameters used in *Tsyganov et al., 2012*. The diffusion coefficients of RhoA and Rac1 were taken from *Das et al., 2015*. The cell shape and size parameters were taken from the imaging data, generated in present study.

## Numerical methods for solving PDE equations

The PDE system (*Equation 10*) was solved numerically by the finite volume method (*Patankar, 1980*) aided by the splitting technique (*Oran and Boris, 1987*), and using the OpenFOAM platform (*Jasak, 2009*). A computational 2D domain was obtained by extracting contours of cells from experimental cell images using the OpenCV library (*Bradski, 2000*) and meshed by non-structured triangular meshes using the Salome platform (*Ribes and Caremoli, 2007*). An example of the

computational mesh is presented in *Figure 3—figure supplement 1*, panel F. The x and y axes were set along the cell length and width as depicted in *Figure 3A*. Distributions of the total concentrations of DIA and ROCK were set according to *Equations 3 and 10*. For equations describing spatio-temporal dynamics of active and inactive forms of Rho and Rac1, zero-gradient boundary conditions were applied. The diffusion term was discretized using unstructured triangular meshes by means of the 'over-relaxed correction' technique (*Jasak, 1996*). ODE systems describing chemical kinetics were solved using fifth-order Cash-Karp embedded Runge-Kutta scheme with error control and adjusted time-step size (*Press et al., 1992*). The simulation results were visualized using the Para-View software package (*Henderson, 2007*).

## Acknowledgements

We thank Dirk Fey (Systems Biology Ireland, University College Dublin) for his input in the modeling and discussions, Kasia Kedziora (Netherlands Cancer Institute) for providing macros for quantifying the PLA images, and Denis Pushin (Moscow Institute of Physics and Technology) for advices on obtaining the contours of cells from the experimental images for carrying out numerical calculations. We thank the Edinburgh Super-Resolution Imaging Consortium for assistance with super-resolution imaging. Supported by NIH/NCI grant R01CA244660, EU grants SmartNanoTox (grant no. 686098), NanoCommons (grant no. 731032), SFI grants 14/IA/2395 and 18/SPP/3522, CRUK Edinburgh Centre C157/A25140, Breast Cancer NOW PR183.

## Additional information

### Funding

| Funder | Grant reference number | Author |
| --- | --- | --- |
| National Institutes of Health | R01CA244660 | Boris N Kholodenko |
| Horizon 2020 Framework Programme | 686098 (SmartNanoTox) | Boris N Kholodenko |
| Horizon 2020 Framework Programme | 731032 (NanoCommons) | Boris N Kholodenko |
| Science Foundation Ireland | 14/IA/2395 | Oleksii S Rukhlenko<br>Walter Kolch |
| Science Foundation Ireland | 18/SPP/3522 | Walter Kolch |
| Cancer Research UK | C157/A25140 (Edinburgh Centre) | Alfonso Bolado-Carrancio<br>Alex von Kriegsheim |
| Breast Cancer Now | PR183 | Alfonso Bolado-Carrancio<br>Alex von Kriegsheim |

The funders had no role in study design, data collection and interpretation, or the decision to submit the work for publication.

### Author contributions

Alfonso Bolado-Carrancio, Elena Nikonova, Anne Wheeler, Amaya Garcia-Munoz, Investigation, Conducted the experiments; Oleksii S Rukhlenko, Investigation, Writing - original draft, Writing - review and editing, Developed the model and took part in the discussion of experiments; Mikhail A Tsyganov, Investigation, Developed the model; Walter Kolch, Writing - original draft, Writing - review and editing, Discussed design of experiments and results; Alex von Kriegsheim, Supervision, Investigation, Designed and conducted experiments; Boris N Kholodenko, Conceptualization, Supervision, Writing - original draft, Writing - review and editing, Conceived the study

### Author ORCIDs

Oleksii S Rukhlenko https://orcid.org/0000-0003-1863-4987
Alex von Kriegsheim https://orcid.org/0000-0002-4952-8573
Boris N Kholodenko https://orcid.org/0000-0002-9483-4975

**Decision letter and Author response**
Decision letter https://doi.org/10.7554/eLife.58165.sa1
Author response https://doi.org/10.7554/eLife.58165.sa2

---

## Additional files

### Supplementary files

• Source code 1. Scripts for building 1D and 2D bifurcation diagrams for ODE system (*Equation 6*). See 'description.txt' file in the zip-archive for details.

• Source code 2. Archive of the OpenFOAM source code for solving reaction-diffusion equations (*Equation 12*). See 'description.txt' file in the zip-archive for details.

• Supplementary file 1. Quantitative mass spectrometry data by *Byrne et al., 2016* used to populate protein abundances in the mathematical model.

• Supplementary file 2. Parameter values (*Equation 12*).

• Transparent reporting form

### Data availability

All data generated or analysed during this study are included in the manuscript and supporting files.

---

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
