## [Decision Letter]

**Acceptance summary:**

We were very impressed with the level of integration of experiment and theory in this study, and the clarity that the authors brought to the murky question of Rac/Rho-regulated cell polarity.

**Decision letter after peer review:**

Thank you for submitting your article "Periodic propagating waves coordinate RhoGTPase network dynamics at the leading and trailing edges during cell migration" for consideration by *eLife*. Your article has been reviewed by three peer reviewers, one of whom is a member of our Board of Reviewing Editors, and the evaluation has been overseen by Aleksandra Walczak as the Senior Editor. The following individual involved in review of your submission has agreed to reveal their identity: Leah Edelstein-Keshet (Reviewer #2).

The reviewers have discussed the reviews with one another and the Reviewing Editor has drafted this decision to help you prepare a revised submission.

The reviewers are enthusiastic about your manuscript.

Essential revisions:

Please focus broadly on the following issues:

1) Please improve the presentation of the study in general, and of the model in particular.

2) Please spell out more clearly the novelty and significance of the findings – what exactly do you consider the new insight into the Rac/Rho signaling system, which comes out of your study.

Reviewer #1:

In this paper, the authors combine mathematical modeling and experiment to investigate how Rac-Rho system self-organizes to regulate migration behaviour of a polarized cell. After establishing that ROCK is more abundant at the cell rear and body, whereas DIA is more abundant at the leading edge, the authors suggest distinct circuitries of RhoA-Rac1 interactions and different RhoA and Rac1 kinetics along a cell. Specifically, oscillations of RhoA and Rac1 activities at the leading edge guide local protrusions and retractions, whereas high, stable RhoA activity and low Rac1-GTP at the rear are beneficial for steady retraction. The leading-edge oscillations are shown to create waves that periodically propagate from the front to the rear. The model makes two nontrivial predictions – about hysteresis of RhoA and Rac1 activities upon PAK inhibition and about formation of multiple lamellipodia in ROCK-inhibited cells – that are both observed.

The study is interesting and novel.

Here are some questions and suggestions:

1) How were the parameter values in Supplementary file 2 determined? I understand that the concentrations are from proteomics, but where are the rates from?

2) There has to be at least some explanation of the model in the main text.

3) The claim is that there are oscillations at the leading edge, but surely the oscillations only occur under pretty restrictive conditions in the parameter space? What are these conditions?

4) I am not quite sure how the reaction-diffusion model is set up. Are there some model parameters that are assumed to be varied along the cell length, like Dia and Rock distributions? What are these parameters?

5) In many places in the Results the authors over-interpret the data and basically mix actual results with speculations. For example, in a number of places in the Results, the statements about release of focal adhesions at the rear are made. But there is no data on the adhesions. So, please, move all speculations to the Discussion.

Reviewer #2:

This is a very interesting paper from the lab of Kolch and Kholodenko that is a significant advance beyond an earlier contribution of this group in (Byrne et al., 2016) on Rac-Rho dynamics.

In this paper, the authors combine careful experiments with advanced computational modeling to untangle a signaling network that consists of both positive and negative feedbacks between Rho and Rac via the effectors ROCK, Dia, and PAK. What is beautiful about this system is that it can explain the front-back polarity and Rac or Rho dominated cells, and many more dynamical regimes in cells such as waves that correspond to cycles of protrusion-retraction at the leading edge.

This research is significant not only for understanding the migration of normal motile cells, but also for addressing the pathology of metastatic cells. It is at the usual high level of sophistication of work that comes from this lab.

I think this paper has high priority for publication, and will interest a wide readership. It will definitely be interesting to members of my group, as a significant new way to understand how GTPase dynamics regulates cell motility, and to consider fully deforming cell domains with our computational methods, based on these results.

My only major suggestion is that the authors should mention the fact that waves of these regulators could interact with a deforming cell edge, result in behaviour that is not captured in the static cell shapes simulated here. (For example, we have found that some waves get damped out by the protrusion of a cell edge in simulations with a fully deforming 2D cell domain.)

Reviewer #3:

Before continuing, I'll note that I am not an experimentalist and as such will not comment on experimental methods.

In this article, the authors take a joint theoretical and experimental approach to study how the interactions of Rho GTPases and their effectors generate different phases of cell motion. The main takeaway is that RhoA and Rac1 activity in different parts of the cell is, at least partly, determined by the compartmentalized localization of ROCK and DIA. In short, DIA is enriched in the leading edge membrane and ROCK in the trailing edge membrane, and these localizations cause different functional GTPase dynamics in the front and rear (e.g. oscillatory front versus a steady state rear).

This is a solid article, though I do have a few broad concerns. First and foremost, I do not see what the substantial new understanding is here. My reading is that it is mainly adding to our understanding of the specific details of how GTPase + effector interactions influence cell dynamics. While this is a useful contribution, it is something I would expect to see in a disciplinary journal such as MBoC or Biophysical Journal. Is there something more that I'm missing here?

Another, albeit smaller concern, is the article is a difficult read. The writing itself is ok. It is the presentation that I'm having trouble with. There are a lot of model details here and I had trouble putting them in context of the main results of each section. In some cases, it wasn't completely clear what the main result of the section is other than illustrating agreement between a model prediction and experiments. It is also currently difficult to compare the model results with experimental results. In a number of places static model images are compared to experimental kymographs or experimental kymographs are compared to model videos. Is it possible to use a more common presentation formats to compare the model and experimental results? Or is it possible to synopsize some features that you are trying to compare in some measure that can be more easily compared.

Here are some other detailed comments:

1) What is actin doing during this process? You are discussing wave like behaviors here. A significant number of articles over the last 10-20 years have suggested that actin may play a role in the propagation of such waves. Any idea if actin has a role here? The possibility should at least be discussed.

2) What would lead to stable, spatial compartments with different levels of DIA and ROCK? That compartmentalization is absolutely critical to all of the results in this article, but I did not find any discussion of what might lead to that stable compartmentalization. This is important to discuss.

3) Figure 1: In regards to this figure, you state that DIA and ROCK activity are preferentially localized near the leading and trailing edges respectively. What do you make of the observation that both appear to be at high activity levels in the middle compartment? Some discussion of this middle would be helpful.

4) Subsection “Spatially variable topology of the RhoA-Rac1 interaction network”, last paragraph: Here you mention that RhoA abundance is higher than that of DIA and ROCK. Are the binding of ROCK and DIA to Rho 1:1 binding proportions. That is, can more than one ROCK (for example) bind to the same Rho?

5) Subsection “Spatiotemporal dynamics of the RhoA-Rac1 network reconciles the distinct temporal behaviors at the cell front and rear”, "… leading to re-arrangement of the cytoskeleton and dissociation of focal adhesions… leads to the rear retraction": This set of text is confusingly mixing model results with model interpretation. You are not modeling focal adhesions or retractions, but currently the text makes it sound as if you are. I suggest, in this area of the text, more clearly delineating what are i) model results, ii) interpretations, and iii) experimental results. At the moment they are a little muddled. I would suggest trying to clarifying this throughout as well.

6) Subsection “Hysteresis of Rac1 and RhoA activities and cell shape features”: I found this section difficult to follow and had a hard time figuring out the message. The text discussing what happens as the system transitions from I  II  III and back is difficult to parse.

7) Subsection “ROCK inhibition results in multiple competing lamellipodia and multi-polar cell shapes”, "… chaotic spontaneous activity bursts.…": Looking at Figure 5B, RhoA activity doesn't look either chaotic or bursty to me. Instead, it looks (to me), like there are a few regions of activity that are somewhat dynamic. Why do you describe it this way?

8) In Figure 5, it would be useful to have kymographs of RhoA activity in both the unperturbed and ROCK inhibited cases, with similar presentation for comparison.

9) I'm having difficulty comparing the kymograph results of Figure 5 with the model simulation video (Video 3, I believe). Is there a more quantitative way to compare these that would support the point you are trying to make? It is just difficult to see the correspondence between the results and your statements about them at the moment.

10) In the last section, you chose to perturb ROCK. Why only ROCK? Up to this point, the discussion had resolved around the importance of ROCK and DIA and their compartmentalization. Why not similarly perturb DIA? This does get mentioned in the Discussion, but why is it not more thoroughly discussed as its own Results section on par with the discussion of ROCK manipulation?

11) Discussion, seventh paragraph: It is worth mentioning (with appropriate references to for example, Orien Weiner, Steve Altschuler, and Adam Cohen) that cell tension is another commonly discussed mechanism coordinating front back signaling.

12) In this model, you include the inactive forms of RhoA and Rac1. However, you also note that complex activity is much more limited by DIA and ROCK amounts. Why include the inactive forms in the model? I guess what I'm really asking here is, does the GTPase conservation play any role here as has been suggested in other past studies? Or is it ROCK and DIA limitations driving everything?

---

## [Author Response]

Essential revisions:Please focus broadly on the following issues:1) Please improve the presentation of the study in general, and of the model in particular.2) Please spell out more clearly the novelty and significance of the findings – what exactly do you consider the new insight into the Rac/Rho signaling system, which comes out of your study.Reviewer #1:[…] Here are some questions and suggestions:1) How were the parameter values in Supplementary file 2 determined? I understand that the concentrations are from proteomics, but where are the rates from?2) There has to be at least some explanation of the model in the main text.

The reviewer’s concern about parameter values is well understood. Unfortunately, performing kinetic measurements is not a hot scientific topic currently. Because many parameters are unknown for any model, we have to make our own measurements and take typical parameter ranges for kinetic constants based on previously developed models and measurements of similar protein interactions and reactions. Many parameters were taken from our previous mechanistic model of the RhoA-Rac1 network, which was tailored to MDA-MB-231 cells (Byrne et al., 2016) – the same cell line we used in this work. As in the previous model, we used quantitative mass spectrometry data to determine the prevailing protein isoforms of the RhoA-Rac1 network in MDA-MB-231 cells, as well as the protein abundances. The parameters of activation and deactivation of RhoA, Rac1 and PAK were estimated based on the literature data (Lyda et al., 2019; Tang et al., 2018). The parameters of activation and deactivation of DIA and ROCK were estimated based on typical association and dissociation constants of protein-protein interactions (Kholodenko et al., 1999). The parameters of hyperbolic multipliers (Equation 5) were estimated based on the parameters used in Tsyganov et al., 2012. The diffusion coefficients of RhoA and Rac1 were taken from Das et al., 2015. The cell shape and size parameters were taken from the imaging data, generated in present study.

We added a paragraph about the selection of parameter values in the Materials and methods section.

3) The claim is that there are oscillations at the leading edge, but surely the oscillations only occur under pretty restrictive conditions in the parameter space? What are these conditions?

We fully agree with the reviewer that for any nonlinear system, oscillations occur only under constrained conditions, and other dynamic behaviors would persist if these conditions are not satisfied. Using a spatially localized, compartmentalized model, we performed parameter scans to reveal the zones in parameter space which correspond to the oscillatory behavior. We present the results showing areas 1 and 3 of self-sustained oscillations in the planes of (1) DIA and ROCK abundances (Figure 2A and Figure 2—figure supplement 1A), (2) PAK and ROCK abundances (Figures 4A and 2C), (3) PAK and DIA abundances (Figure 2—figure supplement 1B). Oscillations are induced due to the Andronov-Hopf bifurcation emerging upon the increase in the DIA abundance. Oscillations disappear due to saddle homoclinic bifurcations upon the further increase in the DIA abundance or the increase in the ROCK abundance. These are bifurcations of co-dimensionality 1. In addition to bifurcations of co-dimensionality 1, oscillations are induced or disappear due to the Bogdanov-Takens bifurcation of co-dimensionality 2.

The areas of different RhoA and Rac1 effector abundances correspond to distinct spatial locations within a cell. Most importantly, area 1 of sustained oscillations and the ‘physiological point’, whose coordinates are (1, 1) in the figures, corresponds to a lower ROCK abundance and higher DIA abundance relative to the cell body. As shown by our experimental data (Figure 1C and D), this area 1 corresponds to the leading edge of a cell. If the ROCK abundance increases markedly or the PAK abundance is substantially reduced, while all other abundances are kept constant, the conditions for oscillations no longer hold. Only at the leading edge, a combination of Rac1 activation by RhoA via DIA and RhoA inhibition by Rac1 via PAK (Figure 2B) results in sustained oscillations of RhoA and Rac1 activities (Figure 2D). In the revised manuscript, we clarified the oscillatory conditions.

4) I am not quite sure how the reaction-diffusion model is set up. Are there some model parameters that are assumed to be varied along the cell length, like Dia and Rock distributions? What are these parameters?

Only the abundances of DIA and ROCK are varied along the cell length (Figure 3B and C), as suggested by our spatially-resolved data (Figure 1A-D). No other parameters are changed along the cell length. In the revised manuscript we further clarify how the reaction-diffusion model is set up by adding the following: ‘The spatial profiles of the total DIA and ROCK concentrations are set by Equation 3 that describes their distribution along the cell length (the x-axis). No other total abundances vary with the cell length in our reaction-diffusion model.’

5) In many places in the results the authors over-interpret the data and basically mix actual results with speculations. For example, in a number of places in the Results, the statements about release of focal adhesions at the rear are made. But there is no data on the adhesions. So, please, move all speculations to the Discussion.

We removed all statements about focal adhesions from the Results, moving speculations to the Discussion.

Reviewer #2:[…] My only major suggestion is that the authors should mention the fact that waves of these regulators could interact with a deforming cell edge, result in behaviour that is not captured in the static cell shapes simulated here. (For example, we have found that some waves get damped out by the protrusion of a cell edge in simulations with a fully deforming 2D cell domain.)

We agree with the reviewer and added in the Discussion that during migration cells continuously change their shapes by making protrusions and retractions. In the revised manuscript, we mention that wave interactions with deforming cell boundaries will likely modulate the propagation patterns (Cao et al., 2019).

Reviewer #3:[…] This is a solid article, though I do have a few broad concerns. First and foremost, I do not see what the substantial new understanding is here. My reading is that it is mainly adding to our understanding of the specific details of how GTPase + effector interactions influence cell dynamics. While this is a useful contribution, it is something I would expect to see in a disciplinary journal such as MBoC or Biophysical Journal. Is there something more that I'm missing here?

We apologize for an apparently muted presentation of the novelty and significance of the findings that confused reviewer #3. There are two main novelties. One is that our model presents a minimal biochemical machinery of the RhoA – Rac1 signaling system that is both necessary and sufficient to explain the different dynamic regimes of Rac and Rho at the leading and trailing edge of migrating cells. There is a vast literature on dynamics of Rac and Rho and how they influence cell migration, but no overarching concept of what features are required for successful cell movement and what are the minimal conditions to generate them. The second advance is that our model also explains how the dramatically different kinetics of the RhoA – Rac1 activities at the trailing and leading edges of the cell are coordinated. This coordination is required for mesenchymal cell migration. In the revised manuscript, we emphasize these advances more clearly.

Interestingly, the first minimal heart oscillator model was purely electrochemical (Noble, 1962), although the heart rhythms are affected by multiple factors. In the past more than 50 years, this minimal model was expanded to understand how various stimuli and mechanical tensions are integrated and influence heart rhythms, and it paved the way for modern computational models and tools, such as Virtual Heart (Noble, 2007). Here we present a minimal biochemical mechanism that guides the cyclic process of cell migration. The model rationalizes how the amoeboid and mesenchymal types of migration interchange by suppression or over-activation of specific RhoA and Rac1 effectors.

Our minimal reaction-diffusion model includes only 5 proteins. Ever increasing pace of acquiring new data about small GTPase, the kinetics and spatial localization of GEFs and GAPs, will definitely help researches generate more complex models to quantitatively understand cell motility and the pathology of metastatic cell migration. Interestingly, the first model of cell cycle in *Xenopus* oocytes by Tyson and Novak (Novak and Tyson, 1993) correctly recognized the necessity of two positive feedback loops in which active CDC2-cyclin dimers activated their own production by activating Cdc25 and inhibiting Wee1. These positive feedbacks generated switch like transitions and because of negative feedback where the CDC2-cyclin stimulated its own distraction, this simple network of 5 proteins (the same protein number is our spatially resolved model) generated autonomous oscillations of cell cycle in *Xenopus* oocytes. Much more sophisticated models of cell cycle in mammalian cells that accounted for multiple cyclins, CDC kinases and numerous control points, such the RB-restriction point were later developed but a minimal generator of autonomous oscillations remained the same. Likewise, the dynamic coordination of dramatically different kinetics of the RhoA – Rac1 activities by periodic propagating waves revealed by our model will be a feature of more detail models.

Another, albeit smaller concern, is the article is a difficult read. The writing itself is ok. It is the presentation that I'm having trouble with.

We substantially changed the presentation.

There are a lot of model details here and I had trouble putting them in context of the main results of each section. In some cases, it wasn't completely clear what the main result of the section is other than illustrating agreement between a model prediction and experiments.

As explained above, a major novelty of our model is that it can explain which Rac – Rho dynamics are necessary and sufficient for successful cell movement. Substantiating this claim requires a systematic analysis of the model and its predictions. Such an analysis is intrinsically detailed. Models are useful abstractions that distil higher order principles out of (in this case experimental) observations that allow us to make predictions about future behavior. Testing the validity of these predictions requires further experiments. This iteration of observation – model building – prediction testing is commonly viewed as the gold standard scientific methodology in systems biology, and we apologize if this has confused the reviewer. As part of the improvements in presentation we also have clarified the principles of the general methodological approach.

It is also currently difficult to compare the model results with experimental results. In a number of places static model images are compared to experimental kymographs or experimental kymographs are compared to model videos. Is it possible to use a more common presentation formats to compare the model and experimental results? Or is it possible to synopsize some features that you are trying to compare in some measure that can be more easily compared.

The model is an ODE model and therefore inherently calculates continuous dynamic processes. To account for this, we have included videos in the supplementary data. Unfortunately, even live imaging methods do not allow the recording of experimental data at similar continuous rates, and most experimental methods only provide snapshots. Therefore, a unified way to present the data is difficult and actually not sensible as it would be misleading. For instance, to illustrate steady state conditions what the reviewer calls “static model images” are adequate. Kymographs are commonly used to illustrate the dynamics of processes in discontinuous experimental data. To illustrate salient model dynamics we have provided videos. Snapshots showing critical phases of these videos allow comparisons with the kymographs. We added kymograph based on modeling data (Figure 3—figure supplement 1C) and clarified the critically important features that are captured both by our model and experiments.

Here are some other detailed comments.1) What is actin doing during this process? You are discussing wave like behaviors here. A significant number of articles over the last 10-20 years have suggested that actin may play a role in the propagation of such waves. Any idea if actin has a role here? The possibility should at least be discussed.

The point is well taken. Downstream effectors of RhoA and Rac1, including ROCK. PAK and DIA included in our model, are all regulators of the cytoskeleton. Feedback and some feedforward loops in our models, such as RhoA-DIA-Rac1; RhoA-ROCK-Rac1 and Rac1-PAK-RhoA are also mediated by actin and myosin dynamics. The spatiotemporal dynamical patterns observed in our study are directly translated into the actin cytoskeleton behavior via downstream effectors of RhoGTPases. We now discuss that actin likely plays a role in the propagation of RhoGTPase waves.

2) What would lead to stable, spatial compartments with different levels of DIA and ROCK? That compartmentalization is absolutely critical to all of the results in this article, but I did not find any discussion of what might lead to that stable compartmentalization. This is important to discuss.

It is suggested that spatial compartmentalization of multiple RhoGTPase-related proteins, including effectors, occurs via interactions with components of the cytoskeleton, such as microtubules (Bartolini and Gundersen, 2010; Liu and Dwyer, 2014; Lovelace et al., 2017; Ren et al., 1998; Takesono et al., 2010).

In the manuscript, we discuss this in the following paragraph: “In addition to diffusion and excitable properties of signaling networks, the cell front and rear can communicate via other molecular mechanisms. […] Different spatial concentration profiles of RhoA and Rac1 downstream effectors considered in our model can depend on the microtubule network.”

3) Figure 1: In regards to this figure, you state that DIA and ROCK activity are preferentially localized near the leading and trailing edges respectively. What do you make of the observation that both appear to be at high activity levels in the middle compartment? Some discussion of this middle would be helpful.

Both ROCK and DIA have functions distinct from cell motility and are known to regulate cytokinetic networks during mitosis as well as the nuclear shape and integrity. The perinuclear pools of DIA and ROCK are responsible for these functions, explaining the co-localization of both effectors in the “middle” region. To simplify the model, we assumed that the plasma membrane DIA and ROCK pools involved in cell migration can be considered independently of perinuclear DIA and ROCK pools. Therefore, in the model (see Figure 3B, C) we describe DIA and ROCK concentrations as given by the intensities of RhoA–DIA and RhoA-ROCK interactions that we determined experimentally (Figure 1).

4) Subsection “Spatially variable topology of the RhoA-Rac1 interaction network”, last paragraph: Here you mention that RhoA abundance is higher than that of DIA and ROCK. Are the binding of ROCK and DIA to Rho 1:1 binding proportions. That is, can more than one ROCK (for example) bind to the same Rho?

RhoA-GTP and Rac1-GTP can only bind one effector molecule at any given time. This is proven by extensive biochemical and structural analyses (Takai et al., 2001).

5) Subsection “Spatiotemporal dynamics of the RhoA-Rac1 network reconciles the distinct temporal behaviors at the cell front and rear”, "… leading to re-arrangement of the cytoskeleton and dissociation of focal adhesions… leads to the rear retraction": This set of text is confusingly mixing model results with model interpretation. You are not modeling focal adhesions or retractions, but currently the text makes it sound as if you are. I suggest, in this area of the text, more clearly delineating what are i) model results, ii) interpretations, and iii) experimental results. At the moment they are a little muddled. I would suggest trying to clarifying this throughout as well.

We moved all statements about focal adhesions into Discussion.

6) Subsection “Hysteresis of Rac1 and RhoA activities and cell shape features”: I found this section difficult to follow and had a hard time figuring out the message. The text discussing what happens as the system transitions from I  II  III and back is difficult to parse.

The system transitions from points I  II  III and back are brought about by varying the PAK abundance, a system parameter. A hallmark of bistable system is hysteresis which is observed by varying a parameter up and down, meaning increasing and then decreasing the parameter. When the parameter increases and reaches a threshold value, the saddle-node bifurcation emerges and a bistable system jumps to the alternative steady state. When the parameter decreases, the system still remains in this alternative steady state when the parameter can have even its initial value until the second saddle-node bifurcation emerges for the reduced parameter value, and the system jumps back to the original steady state. Varying a key parameter both up and down, such as cyclin abundance or PAK abundance, is a classic experimental strategy to detect hysteresis (Pomerening et al., 2003; Sha et al., 2003), and we used it in our previous study (Byrne et al., 2016). In the current paper, we showed a novel, never reported, phenomena of hysteresis experimentally observed in the context of a spatially resolved dynamic system. Moving from point I to point III the system remains in the low RhoA-GTP and high Rac1-GTP state, which is oscillatory in regime 3 and steady state in regimes 7 and 8, until after reaching point II a saddle-node bifurcation shifts the system to the alternative high RhoA-GTP and low Rac1-GTP alternative steady state. In the transition back, from point III to point I the system will remain in high RhoA-GTP and low Rac1-GTP steady state moving through bistable regions until reaching the BiDR region 3 where the Rac1 activity jumps to a high value, whereas the RhoA activity switches to a low value, approaching initial point I. Thus, in our reaction diffusion system dynamical regimes 3, 7 and 8 are critical for observing hysteresis of Rac1 and RhoA activities, which are averaged over time and cell volume as in Western blot experiments in our previous study (Byrne et al., 2016). In the revised manuscript, we further clarified this section (we note that the originally submitted variant of this section was well understood by two other reviewers).

7) Subsection “ROCK inhibition results in multiple competing lamellipodia and multi-polar cell shapes”, "… chaotic spontaneous activity bursts.…": Looking at Figure 5B, RhoA activity doesn't look either chaotic or bursty to me. Instead, it looks (to me), like there are a few regions of activity that are somewhat dynamic. Why do you describe it this way?

As suggested, we deleted any mentioning of ‘chaotic or bursty’ in that piece. It now reads as follows: ‘Measured using the RhoA-GTP FRET-probe, patterns of the RhoA activity (Figure 5B) showed existence of several centers of RhoA activity, whose dynamic behaviors were not coordinated.’

8) In Figure 5, it would be useful to have kymographs of RhoA activity in both the unperturbed and ROCK inhibited cases, with similar presentation for comparison.

In the revised manuscript, the kymographs of RhoA activity in unperturbed cells are presented in Figure 3H, I and Figure 3—figure supplement 1C and D, and the kymograph of RhoA activity in cells treated with ROCK inhibitor is presented in Figure 5B.

9) I'm having difficulty comparing the kymograph results of Figure 5 with the model simulation video (Video 3, I believe). Is there a more quantitative way to compare these that would support the point you are trying to make? It is just difficult to see the correspondence between the results and your statements about them at the moment.

We have quantified number of RhoA-GTP bursts at the leading edge and the cell rear in both our simulations (Figure 5—figure supplement 1B) and the experiments (Figure 5—figure supplement 1C). These figures demonstrate that when ROCK is inhibited, the number of RhoA-GTP bursts does not change at the leading edge, but changes at the cell rear. We describe these results in the section “ROCK inhibition results in multiple competing lamellipodia and multi-polar cell shapes”.

10) In the last section, you chose to perturb ROCK. Why only ROCK? Up to this point, the discussion had resolved around the importance of ROCK and DIA and their compartmentalization. Why not similarly perturb DIA? This does get mentioned in the Discussion, but why is it not more thoroughly discussed as its own Results section on par with the discussion of ROCK manipulation?

ROCK and PAK are kinases, whereas DIA is a formin, involved in the actin polymerization, and it is also a Rho-GTPase effector protein. Several highly specific and fast acting small molecule inhibitors have been developed and characterized for ROCK and PAK, similar to many other kinases. In contrast, for DIA only a few formin inhibitors have been developed (Isogai et al., 2015; Lash et al., 2013). Their specificity is poorly characterized and they have been demonstrated to affect major alternative pathways (such as p53). Due to these off-targets, these inhibitor molecules are unfortunately very cytotoxic. Because of their toxicity and the numerous off-targets that regulate the cytoskeleton independently of DIA they cannot be used in our study. It would be impossible to distinguish the effects that occur due to DIA inhibition from the effects of other affected pathways, which also regulate the cytoskeleton. Thus, siRNA knockdown or CRISPR/Cas9 knockout remain the only tools to study effects of DIA perturbation. However, in both knockdown and knockout experiments, cells adapt their signaling by changing the abundances of multiple proteins in cellular networks. It was shown that following a single knockdown, the entire signaling network gets rewired, known as a butterfly effect (Hart et al., 2015). In our experiments we observed that DIA1 siRNA knockdown results in the substantial changes in the RhoA and Rac1 abundances (Figure 1—figure supplement 1). CRISPR/Cas9 knockout of DIA1 results in similar effects, substantially changing Rac1 abundance, please see the blot in Author response image 1. These data suggest that upon DIA1 knockdown and knockout the entire RhoA-Rac1 network adapts to the new DIA abundance. Because of the absence of experimental tools for rapid DIA activity perturbations, we were constrained to only using rapid PAK and ROCK abundance perturbations.

**Author response image 1. sa2fig1:** 

11) Discussion, seventh paragraph: It is worth mentioning (with appropriate references to for example, Orien Weiner, Steve Altschuler, and Adam Cohen) that cell tension is another commonly discussed mechanism coordinating front back signaling.

As suggested by the reviewer, we now discuss these issues in the revised manuscript. We cite additional references including those mentioned by the reviewer, which describe how physical forces, such as mechanical tension, facilitate coordination of front and back signaling (Houk et al., 2012; Saha et al., 2018; Shi et al., 2018; Wang et al., 2013).

12) In this model, you include the inactive forms of RhoA and Rac1. However, you also note that complex activity is much more limited by DIA and ROCK amounts. Why include the inactive forms in the model? I guess what I'm really asking here is, does the GTPase conservation play any role here as has been suggested in other past studies? Or is it ROCK and DIA limitations driving everything?

GTPases cycle between active and inactive forms. When the active forms change, as for instance during the oscillations, the inactive forms also change. Therefore, the ODE modeling of the Rho GTPase family dynamics requires consideration of both active and inactive forms (Equation 4). Likewise, modeling of the spatiotemporal GTPase dynamics requires consideration of both active and inactive forms, even if the diffusion coefficients of both forms are identical (see reaction-diffusion Equations 9 and 12, which must take into consideration both active and inactive forms in order to describe non-stationary spatiotemporal dynamics of the RhoGTPases. The ratio between the abundances of the RhoGTPases and their effectors does not affect the necessity to consider both active and inactive GTPase forms in a dynamic ODE model.

References:

1) Agladze, K., Aliev, R.R., Yamaguchi, T., and Yoshikawa, K. (1996). Chemical Diode. The Journal of Physical Chemistry 100, 13895-13897.

2) Bartolini, F., and Gundersen, G.G. (2010). Formins and microtubules. Biochim Biophys Acta 1803, 164-173.

3) Hart, J.R., Zhang, Y., Liao, L., Ueno, L., Du, L., Jonkers, M., Yates, J.R., 3rd, and Vogt, P.K. (2015). The butterfly effect in cancer: a single base mutation can remodel the cell. Proc Natl Acad Sci U S A 112, 1131-1136.

4) Isogai, T., van der Kammen, R., and Innocenti, M. (2015). SMIFH2 has effects on Formins and p53 that perturb the cell cytoskeleton. Scientific Reports 5, 9802.

5) Lash, L.L., Wallar, B.J., Turner, J.D., Vroegop, S.M., Kilkuskie, R.E., Kitchen-Goosen, S.M., Xu, H.E., and Alberts, A.S. (2013). Small-Molecule Intramimics of Formin Autoinhibition: A New Strategy to Target the Cytoskeletal Remodeling Machinery in Cancer Cells. Cancer Research 73, 6793.

6) Liu, G., and Dwyer, T. (2014). Microtubule dynamics in axon guidance. Neurosci Bull 30, 569-583.

7) Pomerening, J.R., Sontag, E.D., and Ferrell, J.E. (2003). Building a cell cycle oscillator: hysteresis and bistability in the activation of Cdc2. Nature Cell Biology 5, 346-351.

8) Takai, Y., Sasaki, T., and Matozaki, T. (2001). Small GTP-binding proteins. Physiol Rev 81, 153-208.

9) Takesono, A., Heasman, S.J., Wojciak-Stothard, B., Garg, R., and Ridley, A.J. (2010). Microtubules regulate migratory polarity through Rho/ROCK signaling in T cells. PLoS One 5, e8774.

10) Tsyganov, M.A., Kolch, W., and Kholodenko, B.N. (2012). The topology design principles that determine the spatiotemporal dynamics of G-protein cascades. Mol Biosyst 8, 730-743.